# A UNIFIED FRAMEWORK FOR CONSISTENCY GENERATIVE MODELING

## ABSTRACT

Consistency modeling, a novel generative paradigm inspired by diffusion models, has gained traction for its capacity to facilitate real-time generation through single-step sampling. While its advantages are evident, the understanding of its underlying principles and effective algorithmic enhancements remain elusive. In response, we present a unified framework for consistency generative modeling, without resorting to the predefined diffusion process. Instead, it directly constructs a probability density path that bridges the two distributions. Building upon this novel perspective, we introduce a more general consistency training objective that encapsulates previous consistency models and paves the way for innovative, consistency generation techniques. In particular, we introduce two novel models: Poisson Consistency Models (PCMs) and Coupling Consistency Models (CCMs), which extend the prior distribution of latent variables beyond the Gaussian form. This extension significantly augments the flexibility of generative modeling. Furthermore, we harness the principles of Optimal Transport (OT) to mitigate variance during consistency training, substantially improving convergence and generative quality. Extensive experiments on the generation of synthetic and real-world datasets, as well as image-to-image translation tasks (I2I), demonstrate the effectiveness of the proposed approaches.

## 1 INTRODUCTION

Generative models represent a significant category of machine learning techniques with the capacity to approximate and generate samples from complex, unknown probability distributions. Particularly, the realm of deep generative models has exhibited exceptional advancements across a wide array of domains, such as the generation of images (Dhariwal & Nichol, 2021; Ramesh et al., 2022), speeches (Lu et al., 2022b; Popov et al., 2021), 3D assets (Poole et al., 2022), molecules (Jing et al., 2022; Xu et al., 2022a) and proteins (Liu et al., 2023; Yim et al., 2023). Recent advancements primarily stem from the utilization of the diffusion model framework (Ho et al., 2020; Song et al., 2020b), where the underlying principle involves progressively recovering samples from noise by solving reverse-time stochastic or ordinary differential equations (SDEs/ODEs). This intricate procedure facilitates the extraction of samples that adhere to a desired target data distribution. However, the remarkable outcomes achieved through these methods come at a cost of iterative refinement, often necessitating 10 to 2000 iterations (Song et al., 2020a;b; Lu et al.,

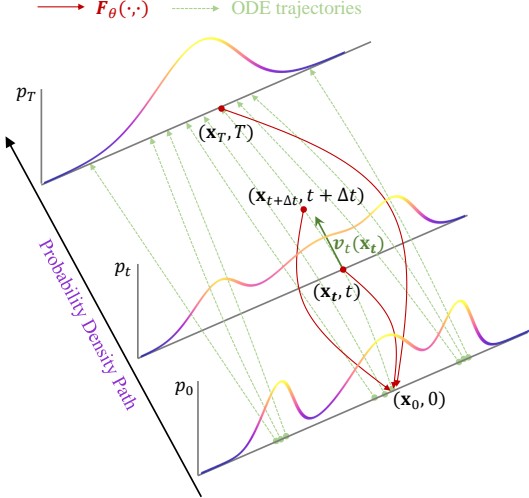

Figure 1: Consistency generative modeling relies on a *probability density path* $\{p_t\}_{t=0}^T$ bridging the prior and data distribution. By collecting two points (e.g., $\mathbf{x}_t$ and $\mathbf{x}_{t+\Delta t} \approx \mathbf{x}_t + \boldsymbol{v}_t(\mathbf{x}_t) \cdot \Delta t$) located on the same trajectory within this path, the network are trained to map them to the initial point (e.g., $\mathbf{x}_0$) for ensuring *self-consistency*.

2022a; Karras et al., 2022) utilizing parameterized score networks. This constraint imposes limitations, particularly concerning real-time applications (Ajay et al., 2022).

To address this challenge, the consistency modeling emerge as a promising paradigm. Specifically, Song et al. (2023) have observed that points existing on the identical solution trajectory of the reverse-time ODE within diffusion models consistently correspond to the same initial point, giving rise to a property termed *self-consistency*. Leveraging this key insight, consistency models harness the capabilities of neural networks to directly establish a mapping between points at arbitrary time instances and their corresponding positions at the initial moment of the trajectory. The training process of these networks is designed to preserve this *self-consistency* property. Song et al. (2023) propose two distinct approaches for determining the two specific points situated along a given trajectory. The first approach involves distilling essential information from a pre-trained score (Liu et al., 2016; Song & Ermon, 2019) network within diffusion models. Conversely, the second approach endeavors to obviate the requirement for a pre-trained score network, instead opting for an unbiased estimation of the score function. This paper primarily centers on the latter approach, as it embodies an independent and self-reliant paradigm for generative modeling.

We argue that, despite consistency models offering the advantage of generating samples in a single step, they are not without limitations. One primary drawback is the constraint imposed on the prior distribution of latent variables in the consistency model, which is restricted to a Gaussian distribution due to its origin from a diffusion process. This constraint inherently restricts the flexibility of generative modeling and its potential applications. Another limitation arises from the estimation of the score function, which, although unbiased, can suffer from high variance. This high variance adversely affects the performance of the consistency model trained using this approach.

To extend the idea of *self-consistency* and unlock the broader potential of consistency models, we present a more general framework for consistency generative modeling through the lens of the *continuity equation* (Benamou & Brenier, 2000). In particular, we identify the foundational components for consistency generative modeling: the *probability density path* connecting the prior and data distributions, and the associated velocity *vector fields* characterizing the movement of particles within this path. The relationship between these components is briefly formulated as the *continuity equation*, a partial differential equation capturing the dynamic evolution of probability density over time. By manipulating these components, such as designing diverse probability density paths and vector fields, we enable the creation of novel consistency models. This key idea eschews the conventional definition involving the reversal of a forward diffusion process (Song et al., 2020b; 2023), which is conceptually simpler and opens up the possibility of exploring novel approaches to consistency modeling.

The contributions of our work are highlighted as follows: **(1)** We provide a unified framework and training objective to understand and explore consistency generation modeling. We demonstrate that our framework encapsulates and improves upon previous methods; **(2)** Building upon the established framework, we introduce novel consistent models for generation: the Poisson Consistency Model (PCM) and the Coupling Consistency Model (CCM). The PCM draws inspiration from electrostatics and exhibits robustness to step size of ODE solver, and the CCM liberates the reliance on simple priors, enabling the facilitation of single-step generation from diverse source distributions; **(3)** We also incorporate Optimal Transport (OT) (Villani et al., 2009) into our framework, demonstrating its efficacy in reducing variance and instability during training. This integration leads to accelerated training convergence and substantial enhancements in performance.

We performed comprehensive experiments on both synthetic and real-world datasets. These experiments encompassed tasks such as unconditional generation of CIFAR-10/CelebA and unpaired image-to-image translation (I2I) using AFHQ. The obtained experimental results affirm the robustness of our consistency model and highlight the substantial improvements achieved by incorporating our framework. Our approach displays promising advancements in single-step generation compared to prior methods.

## 2 PRELIMINARIES

Consistency models are heavily drawn inspiration from the *self-consistency* property found in diffusion models. In this section, we present preliminaries on diffusion models and consistency models.

## 2.1 DIFFUSION MODELS

Diffusion models (Ho et al., 2020; Song et al., 2020b) firstly define a diffusion process $\{\mathbf{x}_t\}_{t=0}^T$ that gradually perturbs the sample $\mathbf{x} \in \mathbb{R}^D$ from $p_{\text{data}}(\mathbf{x})$ into noise,

$$\mathrm{d}\mathbf{x} = \boldsymbol{f}(\mathbf{x}, t)\,\mathrm{d}t + g(t)\mathrm{d}\mathbf{w} \tag{1}$$

where $\boldsymbol{f}(\cdot, \cdot) : \mathbb{R}^D \times [0, T] \to \mathbb{R}^D$ and $g(\cdot) : [0, T] \to \mathbb{R}$ are the drift and diffusion coefficients, $\mathbf{w}$ is the standard Wiener process. The marginal probability density of the variable $\mathbf{x}_t$ is denoted $p_t(\mathbf{x})$ and we clearly have $p_0(\mathbf{x}) \equiv p_{\text{data}}(\mathbf{x})$ by definition. For some specially chosen $\boldsymbol{f}$ and $g$, the transfer kernels $p_{0|t}(\mathbf{x}_t \mid \mathbf{x}_0)$ are of analytic form. Song et al. (2023) adopt the configurations in Karras et al. (2022), setting $\boldsymbol{f}(\mathbf{x}, t) = \mathbf{0}$ and $g(t) = \sqrt{2t}$ so that $p_{0|t}(\mathbf{x}_t \mid \mathbf{x}_0) = \mathcal{N}(\mathbf{x}_0, t^2\boldsymbol{I})$. As $T$ becomes sufficiently large, the distribution of $x_T$ tends towards a tractable Gaussian distribution, i.e., $p_T(\mathbf{x}) \approx \mathcal{N}(\mathbf{0}, T^2\boldsymbol{I})$. In this case, the process in Equation 1 has an associated reverse process that can be described by an ordinary differential equation (ODE), running backward in time and gradually recovering the samples of $p_{\text{data}}(\mathbf{x})$ from the noise:

$$\mathrm{d}\mathbf{x}/\mathrm{d}t = -\nabla \log p_t(\mathbf{x}) \cdot t \tag{2}$$

where $\nabla \log p_t(\mathbf{x})$ represents the score function (Liu et al., 2016). To numerically solve Equation 2, a common approach is to train a neural network using the score matching objective (Vincent, 2011) to approximate $\nabla \log p_t(\mathbf{x})$. However, diffusion models have a bottleneck due to the requirement of iterative access to the neural network, resulting in slower sampling speed.

## 2.2 CONSISTENCY MODELS

To alleviate the aforementioned limitations, a promising solution comes in the form of consistency models (Song et al., 2023). These models are inspired by the remarkable observation that any pair $(\mathbf{x}_t, t)$, which belongs to the same trajectory of the solution to Equation 2, invariably corresponds to the identical initial value $x_0$ at $t = 0$ (i.e. *self-consistency* property). Therefore, the objective of consistency models is precisely defined to preserve this property:

$$\mathcal{L}_{CM}\left(\boldsymbol{\theta}, \boldsymbol{\theta}^-\right) := \mathbb{E}_{t,\mathbf{x}_0,p_{0|t}(\mathbf{x}_t|\mathbf{x}_0)}\left[\lambda(t)d\left(\boldsymbol{F}_\theta\left(\mathbf{x}_{t+\Delta t}, t + \Delta t\right), \boldsymbol{F}_{\theta^-}\left(\mathbf{x}_t, t\right)\right)\right] \tag{3}$$

where $\boldsymbol{F}_\theta(\cdot, \cdot) : \mathbb{R}^D \times [0, T] \to \mathbb{R}^D$ is the parameterized network learning mapping $\mathbf{x}_t$ of any $t \in [0, T]$ to $\mathbf{x}_0$, $\theta^-$ denotes the exponential moving average (EMA) of the network parameters $\theta$ for stabilize the training process, $\lambda(\cdot) : [0, T] \to \mathbb{R}^+$ is a positive weighting function, and $d(\cdot, \cdot) : \mathbb{R}^d \times \mathbb{R}^d \to \mathbb{R}^+$ is the distance function, such as $\ell_2$ distance or Learned Perceptual Image Patch Similarity (LPIPS) (Zhang et al., 2018). To compute $\mathbf{x}_{t+\Delta t}$, Song et al. (2023) propose approximating $\nabla \log p_t(\mathbf{x})$ in Equation 2 as $-(\mathbf{x}_t - \mathbf{x}_0)/t^2$ by Monte Carlo estimation, followed by the application of the Euler solver:

$$\mathbf{x}_{t+\Delta t} = \mathbf{x}_t + [(\mathbf{x}_t - \mathbf{x}_0)/t] \cdot \Delta t, \tag{4}$$

Once Equation 3 is minimized to obtain the optimal parameters $\theta^*$, we can generate samples with a single inference step:

$$\widetilde{\mathbf{x}}_0 = \boldsymbol{F}_{\theta^*}(\mathbf{x}_T, T), \quad \mathbf{x}_T \sim \mathcal{N}(\mathbf{0}, T^2\boldsymbol{I}). \tag{5}$$

Furthermore, consistency models also offer the flexibility of multi-step generation, allowing a trade-off between generative quality and computational consumption. Please refer to the Appendix B.4 for more details.

# 3 A UNIFIED CONSISTENCY GENERATIVE MODELING FRAMEWORK

## 3.1 RETHINKING CONSISTENCY MODELS WITH THE CONTINUITY EQUATION

The right-hand side of Equation 2 actually represents a dynamic time-dependent vector field $\boldsymbol{v}_t(\mathbf{x}) := -\nabla \log p_t(\mathbf{x}) \cdot t$, which elegantly characterizes the velocity of the particle $\mathbf{x}$ at moment $t$. As these particles follow this vector field, the probability density undergoes changes and gives rise to a continuous path $\{p_t(\mathbf{x})\}_{t=0}^T$, which serves as a bridge between the initial distribution $p_0(\mathbf{x}) = p_{\text{data}}(\mathbf{x})$ and the target distribution $p_T(\mathbf{x}) = \mathcal{N}(\mathbf{0}, T^2\boldsymbol{I})$. Thus, this path can seamlessly transform a simple noise distribution into a complex and meaningful data distribution.

An essential property of this process is the conservation of probability mass over the entire sample space and it ensures that samples move by a continuous motion without teleportation. This property leads to the formulation of a *continuity equation*, characterizing the mutual constraints on the density $p_t$ and the vector field $v_t$ :

$$\partial p_t(\mathbf{x})/\partial t = -\nabla \cdot (p_t(\mathbf{x})v_t(\mathbf{x})) \tag{6}$$

where $\nabla \cdot$ represents the divergence operator. We first sample a particle within the path $p_t$ and utilize its $v_t$ to identify adjacent points along the same trajectory. Then, we can train the network that minimizes the output disparity between these points. Building upon this concept, we define a more general consistency objective as:

$$\mathcal{L}_{GCM}\left(\boldsymbol{\theta}, \boldsymbol{\theta}^-\right) := \mathbb{E}_{t,p_t(\mathbf{x})}\left[\lambda(t)d\left(\boldsymbol{F}_\theta\left(\mathbf{x} + v_t(\mathbf{x}) \cdot \Delta t, t + \Delta t\right), \boldsymbol{F}_{\theta^-}\left(\mathbf{x}, t\right)\right)\right] \tag{7}$$

where we opt to use the Euler solver due to its simplicity. By considering different types of probability paths $p_t$ and their corresponding $v_t$, we can construct various models for generative modeling purposes. However, the primary challenge we encounter while training using Equation 7 lies in establishing the $p_t$ that is easy to sample from and the tractable $v_t$, as direct access to them is often unavailable when dealing with high-dimensional distributions of real-world data.

Inspired by recent advances in Flow-based models (Lipman et al., 2022; Tong et al., 2023; Pooladian et al., 2023), we propose a convenient way to bypass the direct handling of $p_t$ and $v_t$. Specifically, we consider leveraging conditional probability density paths $p_t(\mathbf{x} \mid \mathbf{z})$ and its vector fields $v_t(\mathbf{x} \mid \mathbf{z})$ that satisfy Equation 6. We then derive $p_t$ and $v_t$ through marginalization w.r.t a given variable $\mathbf{z}$. Mathematically, it can be expressed as:

$$p_t(\mathbf{x}) = \int p_t(\mathbf{x} \mid \mathbf{z})q(\mathbf{z})d\mathbf{z} \qquad v_t(\mathbf{x}) = \int \frac{v_t(\mathbf{x} \mid \mathbf{z})p_t(\mathbf{x} \mid \mathbf{z})}{p_t(\mathbf{x})}q(\mathbf{z})d\mathbf{z} \tag{8}$$

Remarkably, the marginal $p_t$ and $v_t$ derived from the above equations consistently satisfy Equation 6 (proven in Appendix A.1). This result establishes a significant link between conditional and marginal probability density paths. Consequently, we can decompose complex and intractable probability density paths into simpler conditional forms that rely solely on a variable $\mathbf{z}$. In the following subsection, we instantiate several consistency models by defining specific $p_t(\mathbf{x} \mid \mathbf{z})$ and $v_t(\mathbf{x} \mid \mathbf{z})$.

## 3.2 DIFFUSION CONSISTENCY MODELS

The Diffusion Consistency Model (DCM) introduced by Song et al. (2023) can be viewed as a specialized instance of the framework outlined above. We define the prior distribution $q(\mathbf{z}) = p_{\text{data}}(\mathbf{x})$, and the conditional probability density path as $p_t(\mathbf{x} \mid \mathbf{z}) = \mathcal{N}(\mathbf{z}, t^2\boldsymbol{I}), t \in [0, T]$. Notably, this choice results in a probability path where $p_0(\mathbf{x}) = \int p_0(\mathbf{x} \mid \mathbf{z})q(\mathbf{z})d\mathbf{z} = p_{\text{data}}(\mathbf{x})$, and $p_1(\mathbf{x}) = \int p_1(\mathbf{x} \mid \mathbf{z})q(\mathbf{z})d\mathbf{z} = \mathcal{N}(\mathbf{0}, T^2\boldsymbol{I})$. Utilizing the reparameterization trick (Kingma & Welling, 2013), we can express the samples from $p_t(\mathbf{x} \mid \mathbf{z})$ as $\mathbf{x} = \mathbf{z} + t\epsilon, \epsilon \sim \mathcal{N}(\mathbf{0}, \boldsymbol{I})$. In this context, the closed-form of the conditional vector field $v_t(\mathbf{x} \mid \mathbf{z})$ can be obtained by the following theorem:

**Theorem 1.** *Let $p_t(\mathbf{x} \mid \mathbf{z})$ be a conditional probability path, and the sample from it is denoted as $\mathbf{x} = \beta_t(\mathbf{z}) + \alpha_t(\mathbf{z})\epsilon$, where $\epsilon$ is a random variable independent of $t$. Then, the conditional vector field that induces $p_t(\mathbf{x} \mid \mathbf{z})$ has the form:*

$$v_t\left(\mathbf{x} \mid \mathbf{z}\right) = \frac{d\log\alpha_t\left(\mathbf{z}\right)}{dt}\left(\mathbf{x} - \beta_t\left(\mathbf{z}\right)\right) + \frac{d\beta_t\left(\mathbf{z}\right)}{dt}. \tag{9}$$

The above conclusion extends the Theorem 3 of Lipman et al. (2022) to encompass more general conditions beyond Gaussian probability paths. Plugging the given components into Equation 9 and marginalizing $v_t(\mathbf{x} \mid \mathbf{z})$ by Equation 8, we can derive,

$$v_t(\mathbf{x}) = \int \frac{v_t(\mathbf{x} \mid \mathbf{x}_0)p_t(\mathbf{x} \mid \mathbf{x}_0)}{p_t(\mathbf{x})}p_{\text{data}}(\mathbf{x}_0)d\mathbf{x}_0 = \int \frac{(\mathbf{x} - \mathbf{x}_0)p_t(\mathbf{x} \mid \mathbf{x}_0)}{t\mathbb{E}_{\mathbf{x}_0}[p_t(\mathbf{x} \mid \mathbf{x}_0)]}p_{\text{data}}(\mathbf{x}_0)d\mathbf{x}_0$$
$$= \mathbb{E}_{\mathbf{x}_0 \sim p_{\text{data}}}[\omega(\mathbf{x}_0, \mathbf{x})(\mathbf{x} - \mathbf{x}_0)/t] \tag{10}$$

where $\omega(\mathbf{x}_0, \mathbf{x}) = \exp\left(-\frac{\|\mathbf{x} - \mathbf{x}_0\|_2^2}{2t^2}\right)/\mathbb{E}_{\mathbf{x}_0}[\exp\left(-\frac{\|\mathbf{x} - \mathbf{x}_0\|_2^2}{2t^2}\right)]$. While computing the expectation term mentioned above analytically poses a significant challenge, we can estimate it effectively using Monte Carlo methods. Given a set of samples $\{x_0^i\}_{i=1}^m \sim p_{\text{data}}$, we approximate it as follows,

$$v_t(\mathbf{x}) \approx \sum_{i=1}^{m} \widetilde{\omega}(\mathbf{x}_0^i, \mathbf{x})(\mathbf{x} - \mathbf{x}_0^i)/t \tag{11}$$

Here, $\widetilde{\omega}(\mathbf{x}_0^i, \mathbf{x}) = \exp\left(-\frac{\|\mathbf{x}-\mathbf{x}_0^i\|_2^2}{2t^2}\right) / \sum_{j=1}^m \exp\left(-\frac{\|\mathbf{x}-\mathbf{x}_0^j\|_2^2}{2t^2}\right)$ is an empirical estimation of $\omega$ by self-normalization (Hesterberg, 1995). When employing a single sample for Monte Carlo estimation, the resulting $\boldsymbol{v}_t(\mathbf{x})$ is the same as Equation 4, employed in Song et al. (2023). Nonetheless, the utilization of multiple samples has the potential to significantly reduce the variance in our vector field estimation. We refer the reader to Appendix B.1 for detailed training algorithm of the DCM.

## 3.3 POISSON CONSISTENCY MODELS

The Poisson fields of electrostatics can also offer inspiration for the development of generative models. Xu et al. (2022b) propose an intriguing analogy where $D$-dimensional samples can be viewed as charges situated on the $r = 0$ plane within an augmented $(D+1)$-dimensional space, with $r$ representing the additional dimension. As these charges move along the generated electric field lines, they demonstrate a uniform distribution on a hemisphere with an infinite radius, regardless of the initial charge distribution at $r = 0$. Leveraging this physical analogy, we can design the following novel consistency model. Let us denote the augmented data as $\hat{\mathbf{x}} := (\mathbf{x}, r)$. We introduce the conditional variable $\hat{\mathbf{z}} := (\mathbf{z}, 0) = (\mathbf{x}_0, 0)$, $\mathbf{x}_0 \sim p_{\text{data}}$, and define the conditional probability path $p_t(\hat{\mathbf{x}} \mid \hat{\mathbf{z}})$ as the uniform distribution on a hemisphere centered at $\hat{\mathbf{z}}$ with a radius of $t$. Notably, as the radius grows, any data point can be perceived as lying at the origin of the coordinate system. This property ensures that the $p_t(\hat{\mathbf{x}})$ approximately follows a uniform distribution over the $(D+1)$-dimensional hemisphere for large $t$.

Remarkably, we can employ the approach introduced by Xu et al. (2022b) to replace the anchor variable $t$ with the physically meaningful augmented dimension $r$. Specifically, we get the $r$-dependent conditional probability paths $p_r(\mathbf{x} \mid \mathbf{z})$ by radially projecting a uniform distribution over the surface of a hemisphere onto a hyperplane consisting of points with the same $r$,

$$p_r(\mathbf{x} \mid \mathbf{z}) \propto 1/(\|\mathbf{x} - \mathbf{z}\|_2^2 + r^2)^{\frac{D+1}{2}} \tag{12}$$

In this context, we generate the path $p_r(\mathbf{x} \mid \mathbf{z})$ by constructing samples in the form of $\mathbf{x} = \mathbf{z} + r\sqrt{y/(1-y)}\mathbf{d}_{\text{unit}}$, where $y \sim \text{Beta}(D/2, 1/2)$, and $\mathbf{d}_{\text{unit}}$ is a unit vector sampled from a uniform distribution across angular space. We defer detailed derivations to the Appendix A.3. Considering $r$ as $t$ in Theorem 1, we can obtain the $r$-dependent vector field,

$$\boldsymbol{v}_r(\mathbf{x}) = \eta(\mathbf{x})\mathbb{E}_{\mathbf{x}_0 \sim p_{\text{data}}}\left[\frac{\mathbf{x} - \mathbf{x}_0}{r(\|\mathbf{x} - \mathbf{x}_0\|_2^2 + r^2)^{\frac{D+1}{2}}}\right] \tag{13}$$

where $\eta(\mathbf{x}) = 1/\mathbb{E}_{\mathbf{x}_0}\left[\frac{1}{(\|\mathbf{x}-\mathbf{x}_0\|_2^2+r^2)^{\frac{D+1}{2}}}\right]$. Then, we parameterize a network to learn the mapping of samples within the path to corresponding values on the $r = 0$ plane. The training objective for the Poisson Consistency Model (PCM) can be defined as follows:

$$\mathbb{E}_{r,q(\mathbf{z}),p_r(\mathbf{x}|\mathbf{z})}\left[\lambda(r)d\left(\boldsymbol{F}_\theta\left(\mathbf{x} + \boldsymbol{v}_r(\mathbf{x}) \cdot \Delta r, r + \Delta r\right), \boldsymbol{F}_{\theta^-}\left(\mathbf{x}, r\right)\right)\right] \tag{14}$$

where $q(\mathbf{z}) = p_{\text{data}}(\mathbf{x})$, $r \in [0, r_{\max}]$, and $r_{\max}$ is sufficiently large to ensure $p_{r_{\max}}(\mathbf{x}) \propto 1/(\|\mathbf{x}\|_2^2 + r_{\max}^2)^{\frac{D+1}{2}}$. Once the network is trained, samples can be generated using a single step of inference:

$$\widetilde{\mathbf{x}}_0 = \boldsymbol{F}_{\theta^*}(\mathbf{x}_{r_{\max}}, r_{\max}), \quad \mathbf{x}_{r_{\max}} \sim p_{r_{\max}}. \tag{15}$$

The conditional probability path $p_r(\mathbf{x} \mid \mathbf{z})$ of the PCM shows heavier tails compared to the DCM. As demonstrated by Xu et al. (2023), this characteristic enhances the robustness to estimation errors of the ODE solver, especially in the case of larger step sizes. We present the training algorithm of the PCM in Appendix B.2.

## 3.4 COUPLING CONSISTENCY MODELS

In previous consistency models, the source (prior) distribution has commonly been constrained to a specific simple density, such as Gaussian distribution, which inherently limits the flexibility of generative modeling. To address this constraint, we introduce the Coupling Consistency Model (CCM). Firstly, we define $\mathbf{z}$ as a tuple of random variables, denoted as $\mathbf{z} := (\mathbf{x}_0, \mathbf{x}_1)$, and the distribution $q(\mathbf{z}) := p_{\text{target}}(\mathbf{x}_0)p_{\text{source}}(\mathbf{x}_1)$, where $p_{\text{source}}$ and $p_{\text{target}}$ correspond to the source and target distributions, respectively. We futher introduce a conditional probability density as $p_t(\mathbf{x} \mid$

$\mathbf{z}) = \mathcal{N}(t\mathbf{x}_1 + (1-t)\mathbf{x}_0, \sigma_f^2 \boldsymbol{I})$, representing a linear path as proposed by Lipman et al. (2022). Here, $t \in [0, 1]$ and $\sigma_f$ being a fixed hyperparameter. Clearly, we have the following relationship:

$$p_0(\mathbf{x}) = \int p_0(\mathbf{x} \mid \mathbf{z})q(\mathbf{z})d\mathbf{z} = \int \mathcal{N}(\mathbf{x_0}, \sigma_f^2 \boldsymbol{I})p_{\text{target}}(\mathbf{x}_0)d\mathbf{x}_0 = p_{\text{target}}(\mathbf{x}) \otimes \mathcal{N}(\mathbf{0}, \sigma_f^2 \boldsymbol{I}) \quad (16)$$

$$p_1(\mathbf{x}) = \int p_1(\mathbf{x} \mid \mathbf{z})q(\mathbf{z})d\mathbf{z} = \int \mathcal{N}(\mathbf{x_1}, \sigma_f^2 \boldsymbol{I})p_{\text{source}}(\mathbf{x}_1)d\mathbf{x}_1 = p_{\text{source}}(\mathbf{x}) \otimes \mathcal{N}(\mathbf{0}, \sigma_f^2 \boldsymbol{I}) \quad (17)$$

where $\otimes$ denotes the convolution operation. As $\sigma_f^2 \to 0$, the above $p_t$ can form a path connecting arbitrary two probability densities. Then, we can construct the samples from $p_t(\mathbf{x} \mid \mathbf{z})$ as $\mathbf{x} = t\mathbf{x}_1 + (1-t)\mathbf{x}_0 + \sigma_f^2 \epsilon$, and $\epsilon \sim \mathcal{N}(\mathbf{0}, \boldsymbol{I})$. By applying the results from Theorem 1, we estimate the vector field using the following expression:

$$\boldsymbol{v}_t(\mathbf{x}) = \mathbb{E}_{\mathbf{x}_0, \mathbf{x}_1}[\gamma(\mathbf{x}_0, \mathbf{x}_1, \mathbf{x})(\mathbf{x}_1 - \mathbf{x}_0)] \quad (18)$$

where $\gamma(\mathbf{x}_0, \mathbf{x}_1, \mathbf{x}) = \exp\left(-\frac{\|t\mathbf{x}_1 + (1-t)\mathbf{x}_0 - \mathbf{x}\|_2^2}{2\sigma_f^2}\right) / \mathbb{E}_{\mathbf{x}_0, \mathbf{x}_1}[\exp\left(-\frac{\|t\mathbf{x}_1 + (1-t)\mathbf{x}_0 - \mathbf{x}\|_2^2}{2\sigma_f^2}\right)]$. In practice, for small values of $\sigma_f$, it is expected that most of the $\gamma$ values will be close to zero. Therefore, we estimate the vector field by considering only the $\mathbf{x}_0$ and $\mathbf{x}_1$ used to compute the mean of $p_t(\mathbf{x} \mid \mathbf{z})$. We now delineate the training objectives of the CCM,

$$\mathbb{E}_{t, q(\mathbf{z}), p_t(\mathbf{x}|\mathbf{z})} \left[\lambda(t)d\left(\boldsymbol{F}_\theta\left(\mathbf{x} + (\mathbf{x}_1 - \mathbf{x}_0) \cdot \Delta t, t + \Delta t\right), \boldsymbol{F}_{\theta^-}(\mathbf{x}, t)\right)\right] \quad (19)$$

where $q(\mathbf{z}) = p_{\text{target}}(\mathbf{x}_0)p_{\text{source}}(\mathbf{x}_1)$, $p_t(\mathbf{x} \mid \mathbf{z}) = \mathcal{N}(t\mathbf{x}_1 + (1-t)\mathbf{x}_0, \sigma_f^2 \boldsymbol{I})$. While CCM extends consistency generative modeling to more general source distributions, it is essential to acknowledge that the independent sampling of $\mathbf{x}_0$ and $\mathbf{x}_1$ can lead to high variance in estimating the vector field. To further address this limitation, we have made a noteworthy observation: Equations 16 and 17 remain valid, as long as the joint distributions $\pi(\mathbf{x}_0, \mathbf{x}_1)$ satisfy $\int \pi(\mathbf{x}_0, \mathbf{x}_1)d\mathbf{x}_1 = p_{\text{target}}(\mathbf{x}_0)$ and $\int \pi(\mathbf{x}_0, \mathbf{x}_1)d\mathbf{x}_0 = p_{\text{source}}(\mathbf{x}_1)$. With this in mind, we propose an extension of $q(\mathbf{z})$ to a coupling $\pi(\mathbf{x}_0, \mathbf{x}_1)$ instead of independent distribution, thereby enhancing the model's capabilities. We note that the vector field defined in Equation 19 introduces the difference term $(\mathbf{x}_1 - \mathbf{x}_0)$. Therefore, an intuitive choice for $\pi$ is to leverage the solution of the optimal transport (OT) problem, which aligns the nearest pair $(\mathbf{x}_0, \mathbf{x}_1)$ from the two distributions, akin to (Pooladian et al., 2023), resulting in a lower-variance estimation of $\boldsymbol{v}_t$. Mathematically, the OT problem is formulated as follows,

$$\pi^* = \underset{\pi \in \Pi}{\operatorname{argmin}} \mathbb{E}_{\pi(\mathbf{x}_0, \mathbf{x}_1)}[c(\mathbf{x}_0, \mathbf{x}_1)] \quad (20)$$

Here, $\Pi$ represents the set of coupling probabilities with marginals matching $p_{\text{source}}$ and $p_{\text{target}}$, and $c(\cdot, \cdot) : \mathbb{R}^D \times \mathbb{R}^D \to \mathbb{R}^+$ is employed to quantify the transport cost, with the squared Euclidean distance being our chosen metric. The following theorem highlights the significance of using OT to construct the coupling, enhancing the estimation of the vector field.

**Theorem 2.** *Let $\pi^*$ is the optimal transport plan of Equation 20, the prior $q(\mathbf{z}) := \pi^*$ and the conditional probability density $p_t(\mathbf{x} \mid \mathbf{z}) := \mathcal{N}(t\mathbf{x}_1 + (1-t)\mathbf{x}_0, \sigma_f^2 \boldsymbol{I}), t \in [0, 1]$. Assume $(\mathbf{x}_0, \mathbf{x}_1) \sim q(\mathbf{z})$ and $\mathbf{x} \sim p_t(\mathbf{x} \mid \mathbf{z})$, $\boldsymbol{v}_t(\mathbf{x})$ is the vector field defined in Equation 8. As $\sigma_f \to 0$, we have,*

$$\|(\mathbf{x}_1 - \mathbf{x}_0) - \boldsymbol{v}_t(\mathbf{x})\|_2^2 \to 0 \quad (21)$$

Solving Equation 20 at the distribution level is challenging due to the unavailability of analytical marginal distributions. Instead, we pair samples within a training batch by solving the discrete form of the OT problem, which can be effectively accomplished using the network simplex method (Peyré et al., 2019). For detailed training steps, please refer to the algorithm summarized in Appendix B.

## 4 RELATED WORK

**Diffusion models and consistency models.** The diffusion models (Ho et al., 2020; Song et al., 2020b) serve as a generative model drawing inspiration from thermodynamics. A primary challenge in the diffusion model pertains to the time consumption incurred by the iterative denoising steps that are necessitated. To mitigate this limitation, considerable efforts have been dedicated to expediting the diffusion model, leading to two principal avenues of exploration. The first involves training

an auxiliary network through distillation (Zheng et al., 2023; Luhman & Luhman, 2021; Salimans & Ho, 2022), while the second entails the proposition of advanced numerical solvers or sampling methods (Lu et al., 2022a; Zhang & Chen, 2022). The concept of consistency models (Song et al., 2023) emerged from the consistency property within the probabilistic flow ODE of the reverse diffusion process. They are used for single-step sampling by learning a mapping to an initial point. Recent work by Daras et al. (2023) also delves into harnessing the consistency of reverse-time SDE to augment the diffusion model's capabilities.

**Probability density path modeling.** Chen et al. (2018) introduce an innovative class of continuous-time generative models termed Continuous Normalized Flows (CNFs). These models harness ODEs parameterized by neural networks to generate probability density paths through maximum likelihood training (Grathwohl et al., 2018). However, the training process for such models typically necessitates ODE numerical simulations, leading to computational inefficiencies. Recently, Flow Matching was proposed by Lipman et al. (2022), offering a novel approach by enabling the generation of desired probability density paths without the need for numerical simulations during training. These advances serve as the inspiration for our work. We generalize it into a more versatile form and extend the concept of probability density path modeling to the realm of consistency models.

**OT for generative modeling.** Generative modeling and optimal transport (OT) represent two intimately connected domains. The incorporation of OT regularization has substantially enriched the performance of Generative Adversarial Networks (GANs) and Flow-based models (Yang & Karniadakis, 2020; Onken et al., 2021). Notably, Pooladian et al. (2023) and Tong et al. (2023) harnessed OT techniques for Flow Matching, effectively mitigating the computational expenses associated with sampling. In our work, we use the tools of OT to obtain a coupling, which enhances the vector field estimation in consistency training. In parallel work, De Bortoli et al. (2021) and Chen et al. (2021) delved into the Schrödinger bridge problem, interpreting the diffusion model through the lens of optimal transport with entropy regularization.

## 5 EXPERIMENTS

In this section, we conduct a variety of generative experiments to demonstrate the effectiveness of our proposed models within our framework. First, we faithfully reproduced the Diffusion Consistency Model (DCM) outlined in Song et al. (2023), using the PyTorch (Paszke et al., 2019) library. We also extended the DCM to its multi-sample Monte Carlo variants (DCM-MS). Furthermore, we implemented the Poisson Consistency Model (PCM) and the Coupling Consistency Model (CCM), along with its variant incorporating optimal transport (CCM-OT). We begin with 2D toy experiments (Section 5.1), which illustrate the model's capacity to fit diverse distributions visually. Subsequently, we delve into image generation experiments using the CIFAR-10 (Krizhevsky et al., 2009) and CelebA (Yang et al., 2015) benchmark dataset (Section 5.2). Finally, we evaluate the performance of our framework in handling unpaired image-to-image translation tasks (I2I) on the AFHQ dataset(Choi et al., 2020) (Section 5.3). In practice, we gradually reduce the step size of the Euler solver during training. We refer the readers to the Appendix C for more detailed experimental setups.

### 5.1 2D TOY EXPERIMENTS

In our experiments, we utilize two classic toy datasets: the Swiss roll and two moons datasets (Pedregosa et al., 2011). We define the $d(\cdot, \cdot)$ within the consistency objective as the squared $\ell_2$ distance. During the training phase, we employ a time-dependent fully-connected neural network architecture, comprising three hidden layers with 64 neurons each. To assess the quality of the generated distributions, we compute the Wasserstein-2 distance between these distributions and the ground truth distribution using a dataset of 10k points.

Table 1: Comparison of different models on the 2D dataset.

| Method | Wasserstein-2 distance($\downarrow$) | | |
| --- | --- | --- | --- |
| | swiss roll | two moons | roll-moons |
| DCM | 0.289 | 0.188 | - |
| DCM-MS | 0.247 | 0.155 | - |
| PCM | 0.225 | 0.136 | - |
| CCM | 0.251 | 0.169 | 0.289 |
| CCM-OT | **0.105** | **0.065** | **0.158** |

**Result:** As illustrated in Table 1, both the DCM-MS and PCM models demonstrate a modest improvement over the standard DCM model across all experimental scenarios. Our findings consistently show that the CCM-OT model outperforms all other models, highlighting the efficacy of

incorporating OT principles for generating coupled samples. We also observed similar results in generating moons from roll distribution (Column 3 in Table 1).

Furthermore, we present some visualizations of the generated results in Figure 2. These visual representations depict the ability of CCM-OT to produce distributions that closely resemble the ground truth, underscoring its exceptional capability to capture intricate data patterns. In contrast, the original DCM model appears to struggle, due to the high variance associated with its estimated vector field, resulting in numerous out-of-distribution points.

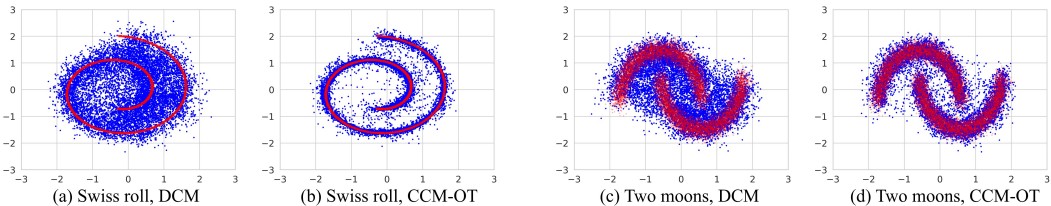

(a) Swiss roll, DCM  (b) Swiss roll, CCM-OT  (c) Two moons, DCM  (d) Two moons, CCM-OT

Figure 2: Randomly generated samples by DCM and CCM-OT, red dots indicate the ground truth data and blue dots indicate generated data.

## 5.2 IMAGE GENERATION

Next, we conduct image generation experiments utilizing the CIFAR-10 and CelebA $64 \times 64$ datasets. In terms of the experimental setup and network architecture, we mostly follow the work of Song et al. (2023). Due to constraints in computational resources, we make adjustments by reducing the batchsize and optimization step. We report two key metrics for evaluation: the Frechet Inception Distance (FID) (Heusel et al., 2017) and the Inception Score (IS) (Salimans et al., 2016). Additionally, we employ the number of function evaluations (NFE) as a measure of inference speed. In particular, we use LPIPS loss as the metric $d(\cdot, \cdot)$ since it performs better on image tasks as outlined in Song et al. (2023).

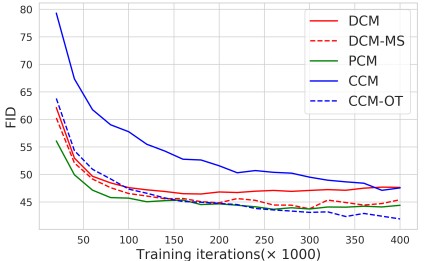

Figure 3: FID on CIFAR-10 throughout training.

**Result:** In Table 2, we report the sample quality of various models on CIFAR-10. We find that the PCM excels in terms of IS. Meanwhile, the CCM-OT model outperforms other consistency models in terms of FID, establishing itself as the leading model for single-step generation. Note that there is a discrepancy between our reported FID and that of Song et al. (2023) for DCM. Our results show a slight decrease compared to Song et al. (2023), and this discrepancy can be attributed to our use of a smaller batch size. However, we emphasize that we have meticulously maintained similar hyperparameters across all models to ensure a fair and equitable comparison. While our models exhibit notable performance, there remains room for improvement, particularly compared to models requiring multiple iterations. We believe that further enhancing our framework's performance can be achieved by increasing the batchsize or exploring advanced network architectures.

Figure 3 illustrates the progression of the FID throughout the training process, computed on 1K samples. It shows that DSM-MS and CCM-OT exhibit faster convergence and ultimately achieve superior performance compared to DSM and CCM. This improvement can be attributed to the more accurate estimation of the vector field, leading to reduced instability in the training. The FID of PCM decreases fastest during the initial training phase, since its heavier-tailed distribution compared to the Gaussian density, which enhances the robustness against the large step size of Euler solvers. These findings align with our theoretical analysis. Additionally, our framework also retains the flexibility of multi-step generation to strike a balance between generation quality and calculation consumption. As reported in Table 3, all models exhibit enhancement as the NFE is increased. With more steps, our proposed approaches still have an advantage on both datasets.

Table 2: Sample quality on CIFAR-10.

| Method | CIFAR-10 | | |
| --- | --- | --- | --- |
| | FID(↓) | IS(↑) | NFE(↓) |
| NCSN (Song & Ermon, 2019) | 25.32 | 8.87 | 1001 |
| DDPM (Ho et al., 2020) | 3.17 | 9.46 | 1000 |
| Score ODE (Song et al., 2020b) | 5.29 | 9.20 | 194 |
| Score SDE (Song et al., 2020b) | 2.20 | **9.89** | 2000 |
| EDM (Karras et al., 2022) | **1.98** | 9.82 | 36 |
| Glow (Kingma & Dhariwal, 2018) | 48.9 | 3.92 | 1 |
| Residual Flow (Chen et al., 2019) | 46.4 | - | 1 |
| GLFlow (Xiao et al., 2019) | 44.6 | - | 1 |
| DenseFlow (Grcić et al., 2021) | 34.9 | - | 1 |
| DCM | 18.4 | 7.15 | 1 |
| DCM-MS | 15.7 | 7.46 | 1 |
| PCM | 14.5 | **8.06** | 1 |
| CCM | 18.3 | 7.40 | 1 |
| CCM-OT | **13.5** | 7.94 | 1 |

Table 3: Effect of NFE with different methods.

| Method | FID(↓) | | |
| --- | --- | --- | --- |
| | NFE = 1 | NFE = 2 | NFE = 5 |
| **CIFAR-10** | | | |
| DCM | 18.4 | 13.6 | 10.9 |
| DCM-MS | 15.7 | **12.1** | 11.2 |
| PCM | 14.5 | 12.6 | **10.2** |
| CCM | 18.3 | 18.5 | 12.9 |
| CCM-OT | **13.5** | 13.7 | 10.6 |
| **CelebA**64 | | | |
| DCM | 41.0 | 27.3 | 25.3 |
| DCM-MS | 38.3 | 22.2 | 18.2 |
| PCM | 32.9 | **17.8** | **16.1** |
| CCM | 38.3 | 34.2 | 30.4 |
| CCM-OT | **30.8** | 28.0 | 25.5 |

### 5.3 UNPAIRED IMAGE-TO-IMAGE TRANSLATION

In this section, we conduct unpaired I2I experiments using the AFHQ dataset to validate the efficacy of our proposed CCM. Specifically, we focus on the task of *Cat→Dog* and *Wild→Dog* translation. All images are uniformly scaled to 256×256. The models are trained with 32 batchsize, and a total of 400K optimization iterations are executed.

**Result:** We present qualitative results generated by CCM-OT in Figure 4. It shows that CCM-OT can preserve domain-independent information such as background, hair color, and posture while modifying domain-specific features like ears, eyes, and noses. We empirically observe that CCM-OT consistently outperforms CCM, achieving better visual quality in terms of authenticity and fidelity (Refer to Figure 8 and Figure 9 in Appendix D). The CCM-OT model enhances the model by effectively aligning similar samples in two domains through OT, rendering it a promising single-step approach for I2I. Note that we directly compute the $\ell_2$ distance for OT in the original pixel space. Exploring the alternative spaces, such as latent spaces based on disentangled representation (Higgins et al., 2016; Sanchez et al., 2020), is a potential avenue for future research.

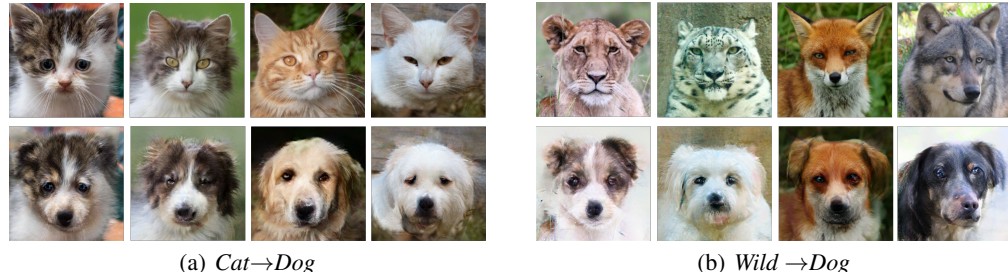

(a) *Cat→Dog*                                        (b) *Wild →Dog*

Figure 4: Translation samples on AFHQ $256 \times 256$ by CCM-OT.

## 6 CONCLUSION

In this paper, we introduce a unified and comprehensive framework for consistency generative modeling. Our framework not only provides insights into existing models but also introduces novel approaches that enhance the versatility and practicality of generative modeling. Additionally, we incorporate OT to improve the performance of these models further. Through empirical experiments, we validate the effectiveness of our proposed framework across a range of generative tasks. Our framework can facilitate single-step generative models, opening up new possibilities for imaginative applications of generative systems. In future research, our aim is to refine and optimize our model to achieve even greater performance gains.

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

# A  PROOFS

## A.1  PROOF THAT MARGINAL PATH AND VECTOR FIELD IN EQUATION 8 SATISFIES THE CONTINUITY EQUATION

Note that $p_t(\mathbf{x} \mid \mathbf{z})$ and $\boldsymbol{v}_t(\mathbf{x} \mid \mathbf{z})$ satisfy the continuity equation, from which we can derive this:

$$\partial p_t(\mathbf{x} \mid \mathbf{z})/\partial t = -\nabla \cdot (p_t(\mathbf{x} \mid \mathbf{z})\boldsymbol{v}_t(\mathbf{x} \mid \mathbf{z})) \tag{22}$$

Then,

$$
\begin{aligned}
\partial p_t(\mathbf{x})/\partial t &= \int (\partial p_t(\mathbf{x} \mid \mathbf{z})/\partial t) q(\mathbf{z}) d\mathbf{z} \\
&= \int (-\nabla \cdot (p_t(\mathbf{x} \mid \mathbf{z})\boldsymbol{v}_t(\mathbf{x} \mid \mathbf{z}))) q(\mathbf{z}) d\mathbf{z} && \text{by Equation 22} \\
&= -\nabla \cdot (\int (p_t(\mathbf{x} \mid \mathbf{z})\boldsymbol{v}_t(\mathbf{x} \mid \mathbf{z})) \, q(\mathbf{z}) d\mathbf{z}) \\
&= -\nabla \cdot (p_t(\mathbf{x}) \int \frac{p_t(\mathbf{x} \mid \mathbf{z})\boldsymbol{v}_t(\mathbf{x} \mid \mathbf{z})}{p_t(\mathbf{x})} q(\mathbf{z}) d\mathbf{z}) \\
&= -\nabla \cdot (p_t(\mathbf{x})\boldsymbol{v}_t(\mathbf{x})) && \text{by the definition of } \boldsymbol{v}_t(\mathbf{x})
\end{aligned}
\tag{23}
$$

The first and third equations above utilize the Leibniz Rule, also known as the product rule for differentiation under the integral sign, allows you to exchange the order of integration and differentiation when certain regularity conditions are satisfied.

## A.2  PROOF OF THEOREM 1

As $\mathbf{x}$ represents samples drawn from the probability path $p_t(\mathbf{x} \mid \mathbf{z})$, which is governed by the conditional vector field $\boldsymbol{v}_t(\mathbf{x} \mid \mathbf{z})$, we can leverage the definition of a dynamic time-dependent vector field to establish the following equation:

$$\frac{d\mathbf{x}}{dt} = \boldsymbol{v}_t(\mathbf{x} \mid \mathbf{z}) \tag{24}$$

Next, by differentiating the expression $\mathbf{x} = \beta_t(\mathbf{z}) + \alpha_t(\mathbf{z})\epsilon$ with respect to time $t$, we obtain:

$$\boldsymbol{v}_t(\mathbf{x} \mid \mathbf{z}) = \frac{d\mathbf{x}}{dt} = \frac{d\beta_t(\mathbf{z})}{dt} + \frac{d\alpha_t(\mathbf{z})}{dt}\epsilon \tag{25}$$

By utilizing the relationship $\mathbf{x} = \beta_t(\mathbf{z}) + \alpha_t(\mathbf{z})\epsilon$, we can express $\epsilon$ in terms of $\mathbf{x}$ and $\alpha_t(\mathbf{z})$ for $\alpha_t(\mathbf{z}) > 0$ as follows:

$$\epsilon = \frac{\mathbf{x} - \beta_t(\mathbf{z})}{\alpha_t(\mathbf{z})} \tag{26}$$

Plugging Equation 26 in Equation 25, we derive the final expression for $\boldsymbol{v}_t(\mathbf{x} \mid \mathbf{z})$:

$$
\begin{aligned}
\boldsymbol{v}_t(\mathbf{x} \mid \mathbf{z}) &= \frac{d\beta_t(\mathbf{z})}{dt} + \frac{d\alpha_t(\mathbf{z})}{dt}\frac{\mathbf{x} - \beta_t(\mathbf{z})}{\alpha_t(\mathbf{z})} \\
&= \frac{d\log\alpha_t(\mathbf{z})}{dt}(\mathbf{x} - \beta_t(\mathbf{z})) + \frac{d\beta_t(\mathbf{z})}{dt}
\end{aligned}
\tag{27}
$$

## A.3  PROOF FOR THE CONDITIONAL PROBABILITY DENSITY PATH W.R.T. $r$ OF PCM

We obtain the $r$-dependent conditional probability density paths by radially projecting a uniform distribution on the surface of a hemisphere onto a hyperplane consisting of points with the same $r$. Specifically, we use the radio of infinitesimal element to estimate $p_r(\mathbf{x} \mid \mathbf{z})$. As depicted in Figure 5, we denote the uniform distribution on the surface of a hemisphere as

$$p_t(\hat{\mathbf{x}} \mid \hat{\mathbf{z}}) = \frac{2}{S_D(1)r^D} \tag{28}$$

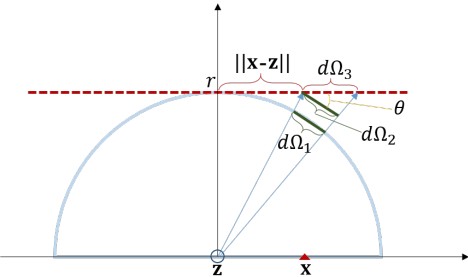

Figure 5: Illustration for calculating conditional density w.r.t. $r$.

where $S_D(1)$ represents the geometric constant for surface area of the unit $D$-sphere. By projecting $d\Omega_1$ onto $d\Omega_3$, we have the following equation for the probability mass,

$$p_t(\hat{\mathbf{x}} \mid \hat{\mathbf{z}})d\Omega_1 = p_r(\mathbf{x} \mid \mathbf{z})d\Omega_3 \tag{29}$$

Based on the projective geometry, we have the following conclusion for the radio of infinitesimal element,

$$\frac{d\Omega_1}{d\Omega_2} = \left(\frac{r}{\sqrt{(||\mathbf{x} - \mathbf{z}||_2^2 + r^2)}}\right)^D \qquad \frac{d\Omega_2}{d\Omega_3} = \cos\theta = \frac{r}{\sqrt{(||\mathbf{x} - \mathbf{z}||_2^2 + r^2)}} \tag{30}$$

Then, we have:

$$\begin{aligned}
p_r(\mathbf{x} \mid \mathbf{z}) &= p_t(\hat{\mathbf{x}} \mid \hat{\mathbf{z}})\frac{d\Omega_1}{d\Omega_3} \\
&= p_t(\hat{\mathbf{x}} \mid \hat{\mathbf{z}})\frac{d\Omega_1}{d\Omega_2}\frac{d\Omega_2}{d\Omega_3} \\
&= \frac{2}{S_D(1)r^D}\left(\frac{r}{\sqrt{(||\mathbf{x} - \mathbf{z}||_2^2 + r^2)}}\right)^D \frac{r}{\sqrt{(||\mathbf{x} - \mathbf{z}||_2^2 + r^2)}} \\
&= \frac{2r}{S_D(1)(\sqrt{(||\mathbf{x} - \mathbf{z}||_2^2 + r^2)})^{D+1}} \\
&\propto 1/(||\mathbf{x} - \mathbf{z}||_2^2 + r^2)^{\frac{D+1}{2}}
\end{aligned} \tag{31}$$

Note that Xu et al. (2023) suggested that the above prior can be further generalized by increasing the number of augmented dimensions, denoted as $K$. Similarly, we can replace the number of augmented dimensions 1 to $K$,

$$p_r(\mathbf{x} \mid \mathbf{z}) \propto 1/(||\mathbf{x} - \mathbf{z}||_2^2 + r^2)^{\frac{D+K}{2}} \tag{32}$$

where $K$ can be used as a hyperparameter to control the degree of heavy-tailedness of $p_r(\mathbf{x} \mid \mathbf{z})$, in our practice of toy and image experiments, we take $K$ to be 5 and 128, respectively, and we find that it works well.

The above distribution can be sampled as follows:

$$\begin{aligned}
\mathbf{d}_{\text{unit}} &= \mathbf{d}/||\mathbf{d}||, \quad \mathbf{d} \sim \mathcal{N}(\mathbf{0}, \mathbf{I}); \\
n &= y/1 - y, y \sim \text{Beta}(D/2, K/2); \\
\mathbf{x} &= (\mathbf{z} + r\sqrt{n}\mathbf{d}_{\text{unit}}) \sim p_r(\mathbf{x} \mid \mathbf{z})
\end{aligned} \tag{33}$$

where the first step samples the unit vector $\mathbf{d}_{\text{unit}}$ from a uniform distribution on the angle, and the second step samples a norm $n$ obeying an inverted Beta distribution for scaling $\mathbf{d}_{\text{unit}}$.

We now prove this conclusion. The density of the inverted Beta distribution is,

$$p(n) \propto n^{\frac{D}{2}-1}(1 + n)^{-\frac{D+K}{2}} \tag{34}$$

And then, by change-of-variable, we have the density for $n' := r\sqrt{n}$,

$$
\begin{aligned}
p\left(n'\right) &\propto n^{\frac{D}{2}-1}\left(1+n\right)^{-\frac{D}{2}-\frac{K}{2}}\frac{2n'}{r^2} \\
&\propto \frac{n'n^{\frac{D}{2}-1}}{(1+n)^{\frac{D+K}{2}}} \\
&= \frac{(n'/r)^{D-1}}{(1+((n')^2/r^2))^{\frac{D+K}{2}}} \\
&\propto \frac{(n')^{D-1}}{(1+((n')^2/r^2))^{\frac{D+K}{2}}} \\
&\propto \frac{(n')^{D-1}}{((n')^2+r^2)^{\frac{D+K}{2}}}
\end{aligned}
\tag{35}
$$

As $n'$ can be interpreted as the norm $\|\mathbf{x} - \mathbf{z}\|_2$, we can express $\mathbf{x}$ as $\mathbf{z} + n'\mathbf{d}_{\text{unit}}$, which follows the distribution $p_r(\mathbf{x} \mid \mathbf{z}) \propto 1/(\|\mathbf{x} - \mathbf{z}\|_2^2 + r^2)^{\frac{D+K}{2}}$.

## A.4 PROOF OF THEOREM 2

Note that $(\mathbf{x}_0, \mathbf{x}_1) \sim q(\mathbf{z}) := \pi^*$ and $\mathbf{x} \sim p_t(\mathbf{x} \mid \mathbf{x}_0, \mathbf{x}_1)$. Since $\pi^*$ is the solution of the OT problem in Equation 20, for any $(\mathbf{x}_0', \mathbf{x}_1') \sim \pi^*$ but is distinct with $(\mathbf{x}_0, \mathbf{x}_1)$, as $\sigma_f \to 0$, we have,

$$
KL(p_t(\mathbf{x} \mid \mathbf{x}_0, \mathbf{x}_1) \| p_t(\mathbf{x} \mid \mathbf{x}_0', \mathbf{x}_1')) \to \infty
\tag{36}
$$

Otherwise, it indicates that the point $(1-t)\mathbf{x}_0 + t\mathbf{x}_1$ coincides with $(1-t)\mathbf{x}_0' + t\mathbf{x}_1'$. According to the trigonometric inequality, we have,

$$
\|\mathbf{x}_0 - \mathbf{x}_1'\|_2 + \|\mathbf{x}_0' - \mathbf{x}_1\|_2 < \|\mathbf{x}_0 - \mathbf{x}_1\|_2 + \|\mathbf{x}_0' - \mathbf{x}_1'\|_2
\tag{37}
$$

The above conclusion clearly contradicts that $\pi^*$ is an optimal transport plan. Referring back to Equation 36, it shows that when $(\mathbf{x}_0, \mathbf{x}_1)$ and $t$ are determined, the values of $x$ are also determined. Thus, we can derive that,

$$
p_t(\mathbf{x}_0, \mathbf{x}_1 \mid \mathbf{x}) = 1 \quad p_t(\mathbf{x}_0', \mathbf{x}_1' \mid \mathbf{x}) = 0
\tag{38}
$$

That is, the distribution $p_t(\mathbf{z} \mid \mathbf{x})$ is a Dirac distribution. From the definition of $\boldsymbol{v}_t(\mathbf{x})$,

$$
\begin{aligned}
\boldsymbol{v}_t(\mathbf{x}) &= \int \frac{\boldsymbol{v}_t(\mathbf{x} \mid \mathbf{z})p_t(\mathbf{x} \mid \mathbf{z})}{p_t(\mathbf{x})}q(\mathbf{z})d\mathbf{z} \\
&= \int \boldsymbol{v}_t(\mathbf{x} \mid \mathbf{z})p_t(\mathbf{z} \mid \mathbf{x})d\mathbf{z} \\
&= \int \boldsymbol{v}_t(\mathbf{x} \mid \mathbf{z})\delta(\mathbf{z} - [\mathbf{x}_0, \mathbf{x}_1])d\mathbf{z} \\
&= \mathbf{x}_1 - \mathbf{x}_0
\end{aligned}
\tag{39}
$$

In summary, as the variance $\sigma_f \to 0$, we have

$$
\|(\mathbf{x}_1 - \mathbf{x}_0) - \boldsymbol{v}_t(\mathbf{x})\|_2^2 \to 0
\tag{40}
$$

# B TRAINING AND SAMPLING DETAILS FOR PROPOSED MODELS

In this section, we will introduce the training algorithms for the four different consistency models described in Sections 3.2 to 3.4 of the main text, i.e., Diffusion Consistency Model (DCM), Poisson Consistency Model (PCM) and Coupling Consistency Model (CCM) and their variants. In addition, we also introduce the multistep sampling algorithm.

## B.1 ALGORITHM FOR DCM AND DCM-MS

Training algorithms for DCM and DCM-MS are presented in Algorithm 1 and Algorithm 2, respectively. In practical applications, we discretize the time horizon $[0, T]$ into $N - 1$ sub-intervals, where the boundaries are defined as $t_0 = 0$ and $t_N = T$. To ensure practical performance, we adopt the step schedule denoted as $N(\cdot)$ and the Exponential Moving Average (EMA) decay rate schedule denoted as $\mu(\cdot)$, as introduced by Song et al. (2023). Here, $\mathcal{U}[\![1, N - 1]\!]$ represents the uniform distribution over the set $1, 2, \cdots, N - 1$, and $d(\cdot, \cdot)$ is a metric function that adheres to the properties $\forall \mathbf{x}, \mathbf{y} : d(\mathbf{x}, \mathbf{y}) \leq 0$ and $d(\mathbf{x}, \mathbf{y}) = 0$ if and only if $\mathbf{x} = \mathbf{y}$. These schedule choices and metric properties play a crucial role in the practical implementation of our proposed algorithms. We highlight in blue where there are differences in the algorithms.

---

**Algorithm 1** Training procedure for DCM

1: **Input:** initial model parameters $\theta$, learning rate $\eta$, step schedule $N(\cdot)$, EMA decay rate schedule $\mu(\cdot)$, metric function $d(\cdot, \cdot)$ and weighting function $\lambda(\cdot)$
2: $\theta^- \leftarrow \theta$ and $k \leftarrow 0$
3: **repeat**
4: $\quad$ Sample $n \sim \mathcal{U}[\![1, N(k) - 1]\!]$ and $\{\mathbf{x}_0^i\}_{i=1}^m \sim p_{\text{data}}$
5: $\quad$ Sample $\{\mathbf{x}^i \sim p_{t_n}(\mathbf{x} \mid \mathbf{x}_0^i)\}_{i=1}^m$
6: $\quad \boldsymbol{v}_{t_n}(\mathbf{x}^i) \leftarrow (\mathbf{x}^i - \mathbf{x}_0^i) / t_n, \forall \mathbf{x}^i$
7: $\quad \mathcal{L}(\theta, \theta^-) \leftarrow \sum_{i=1}^m \lambda(t_n) d(\boldsymbol{F}_\theta(\mathbf{x}^i + \boldsymbol{v}_{t_n}(\mathbf{x}^i) \cdot (t_{n+1} - t_n), t_{n+1}), \boldsymbol{F}_{\theta^-}(\mathbf{x}^i, t_n))$
8: $\quad \theta \leftarrow \theta - \eta \nabla_\theta \mathcal{L}(\theta, \theta^-)$
9: $\quad \theta^- \leftarrow \text{stopgrad}(\mu(k)\theta^- + (1 - \mu(k))\theta)$
10: $\quad k \leftarrow k + 1$
11: **until** convergence

---

**Algorithm 2** Training procedure for DCM-MS

1: **Input:** initial model parameters $\theta$, learning rate $\eta$, step schedule $N(\cdot)$, EMA decay rate schedule $\mu(\cdot)$, metric function $d(\cdot, \cdot)$ and weighting function $\lambda(\cdot)$
2: $\theta^- \leftarrow \theta$ and $k \leftarrow 0$
3: **repeat**
4: $\quad$ Sample $n \sim \mathcal{U}[\![1, N(k) - 1]\!]$ and $\{\mathbf{x}_0^i\}_{i=1}^m \sim p_{\text{data}}$
5: $\quad$ Sample $\{\mathbf{x}^i \sim p_{t_n}(\mathbf{x} \mid \mathbf{x}_0^i)\}_{i=1}^m$
6: $\quad \boldsymbol{v}_{t_n}(\mathbf{x}^i) \leftarrow \sum_{j=1}^m \widetilde{\omega}\left(\mathbf{x}_0^j, \mathbf{x}^i\right)\left(\mathbf{x}^i - \mathbf{x}_0^j\right) / t_n, \forall \mathbf{x}^i$
7: $\quad \mathcal{L}(\theta, \theta^-) \leftarrow \sum_{i=1}^m \lambda(t_n) d(\boldsymbol{F}_\theta(\mathbf{x}^i + \boldsymbol{v}_{t_n}(\mathbf{x}^i) \cdot (t_{n+1} - t_n), t_{n+1}), \boldsymbol{F}_{\theta^-}(\mathbf{x}^i, t_n))$
8: $\quad \theta \leftarrow \theta - \eta \nabla_\theta \mathcal{L}(\theta, \theta^-)$
9: $\quad \theta^- \leftarrow \text{stopgrad}(\mu(k)\theta^- + (1 - \mu(k))\theta)$
10: $\quad k \leftarrow k + 1$
11: **until** convergence

---

## B.2 ALGORITHM FOR PCM

The following algorithm is the training procedure for PCM, and in practice, we divide $[0, r_{\max}]$ into sub-intervals in a similar way as $t$ and generalize the number of augmented dimensions to $K$ as discussed in Appendix A.3. We also use sampling to estimate the vector field,

$$\boldsymbol{v}_r(\mathbf{x}) \approx \widetilde{\eta}(\mathbf{x}) \sum_{i=1}^{m} \left[ \frac{\mathbf{x} - \mathbf{x}_0^i}{r(||\mathbf{x} - \mathbf{x}_0^i||_2^2 + r^2)^{\frac{D+K}{2}}} \right] \tag{41}$$

where $\widetilde{\eta}(\mathbf{x}) = 1 / \sum_{i=1}^{m} \left[ \frac{1}{(||\mathbf{x}-\mathbf{x}_0^i||_2^2+r^2)^{\frac{D+K}{2}}} \right]$ is the empirical estimation of $\eta$.

---

**Algorithm 3** Training procedure for PCM

1: **Input:** initial model parameters $\theta$, learning rate $\eta$, step schedule $N(\cdot)$, EMA decay rate schedule $\mu(\cdot)$, metric function $d(\cdot, \cdot)$ and weighting function $\lambda(\cdot)$
2: $\theta^- \leftarrow \theta$ and $k \leftarrow 0$
3: **repeat**
4:     Sample $n \sim \mathcal{U}[\![1, N(k) - 1]\!]$ and $\{\mathbf{x}_0^i\}_{i=1}^{m} \sim p_{\text{data}}$
5:     Sample $\{\mathbf{x}^i \sim p_{r_n}(\mathbf{x} \mid \mathbf{x}_0^i)\}_{i=1}^{m}$
6:     $\boldsymbol{v}_{r_n}(\mathbf{x}^i) \leftarrow \widetilde{\eta}(\mathbf{x}^i) \sum_{j=1}^{m} \left[ \frac{\mathbf{x}^i - \mathbf{x}_0^j}{r_n(||\mathbf{x}^i - \mathbf{x}_0^j||_2^2 + r^2)^{\frac{D+K}{2}}} \right], \forall \mathbf{x}^i$
7:     $\mathcal{L}(\theta, \theta^-) \leftarrow \sum_{i=1}^{m} \lambda(r_n) d(\boldsymbol{F}_\theta(\mathbf{x}^i + \boldsymbol{v}_{r_n}(\mathbf{x}^i) \cdot (r_{n+1} - r_n), r_{n+1}), \boldsymbol{F}_{\theta^-}(\mathbf{x}^i, r_n))$
8:     $\theta \leftarrow \theta - \eta \nabla_\theta \mathcal{L}(\theta, \theta^-)$
9:     $\theta^- \leftarrow \text{stopgrad}(\mu(k)\theta^- + (1 - \mu(k))\theta)$
10:     $k \leftarrow k + 1$
11: **until** convergence

---

## B.3 ALGORITHM FOR CCM AND CCM-OT

The following algorithm is the training flow for CCM/CM-OT, which differs in whether or not OT is used to generate coupled samples (highlighted in blue), where the calculation of OT can be done using POT library (Flamary et al., 2021) implementation. We denote $\mathcal{U}[\![\{\mathbf{x}_0^i\}_{i=1}^{m}]\!]$ as the uniform distribution w.r.t. each $\mathbf{x}_0^i$.

---

**Algorithm 4** Training procedure for CCM/CCM-OT

1: **Input:** initial model parameters $\theta$, learning rate $\eta$, step schedule $N(\cdot)$, EMA decay rate schedule $\mu(\cdot)$, metric function $d(\cdot, \cdot)$ and weighting function $\lambda(\cdot)$
2: $\theta^- \leftarrow \theta$ and $k \leftarrow 0$
3: **repeat**
4:     Sample $n \sim \mathcal{U}[\![1, N(k) - 1]\!]$ and $\{\mathbf{x}_0^i\}_{i=1}^{m} \sim p_{\text{source}}, \{\mathbf{x}_1^i\}_{i=1}^{m} \sim p_{\text{target}}$
5:     $\pi^* \leftarrow \text{OT}(\mathcal{U}[\![\{\mathbf{x}_0^i\}_{i=1}^{m}]\!], \mathcal{U}[\![\{\mathbf{x}_1^i\}_{i=1}^{m}]\!])$
6:     Sample $\{(\mathbf{x}_0^i, \mathbf{x}_1^i)\}_{i=1}^{m} \sim \pi^*$
7:     Sample $\{\mathbf{x}^i \sim p_{t_n}(\mathbf{x} \mid \mathbf{x}_0^i, \mathbf{x}_1^i)\}_{i=1}^{m}$
8:     $\boldsymbol{v}_{t_n}(\mathbf{x}^i) \leftarrow \mathbf{x}^i - \mathbf{x}_0^i, \forall \mathbf{x}^i$
9:     $\mathcal{L}(\theta, \theta^-) \leftarrow \sum_{i=1}^{m} \lambda(t_n) d(\boldsymbol{F}_\theta(\mathbf{x}^i + \boldsymbol{v}_{t_n}(\mathbf{x}^i) \cdot (t_{n+1} - t_n), t_{n+1}), \boldsymbol{F}_{\theta^-}(\mathbf{x}^i, t_n))$
10:     $\theta \leftarrow \theta - \eta \nabla_\theta \mathcal{L}(\theta, \theta^-)$
11:     $\theta^- \leftarrow \text{stopgrad}(\mu(k)\theta^- + (1 - \mu(k))\theta)$
12:     $k \leftarrow k + 1$
13: **until** convergence

---

### B.4 ALGORITHM FOR MULTI-STEP SAMPLING

The following algorithm, as introduced in Song et al. (2023), initiates at $p_1$ and generates samples from $p_0$ through a multi-step forward process. This algorithm is directly applicable to models such as DCM, DCM-MS, and PCM. For CCM and CCM-OT, a simple modification is required, replacing the 7th line with $\mathbf{x}_{\tau_n} = (1 - \sqrt{\tau_n^2 - \alpha^2})\widetilde{\mathbf{x}}_0 + \sqrt{\tau_n^2 - \alpha^2}\mathbf{x}$.

In our CIFAR-10 generation experiments, for DCM, DCM-MS, CCM, and CCM-OT, we configured the $p_1$ as $\mathcal{N}(\mathbf{0}, \boldsymbol{I})$. However, for PCM, we generated samples from $p_1$ using the following form: $\mathbf{x} = \sqrt{y/(1-y)}\mathbf{d}_{\text{unit}}$.

---

**Algorithm 5** Multi-step sampling

---

1: **Input:** trained consistency model $\boldsymbol{F}_\theta$, sequence of anchor variable, $\tau_1 > \tau_2 > \cdots > \tau_N$, the distributions $p_1$ accessible by samples, $\alpha$ is a small value for avoiding numerical instability
2: $\mathbf{x} \sim p_1$
3: $\mathbf{x}_{\tau_1} \leftarrow \tau_1 \mathbf{x}$
4: $\widetilde{\mathbf{x}}_0 \leftarrow \boldsymbol{F}_\theta(\mathbf{x}_{\tau_1}, \tau_1)$
5: **for** $n = 1$ to $N$ **do**
6: $\quad \mathbf{x} \sim p_1$
7: $\quad \mathbf{x}_{\tau_n} \leftarrow \widetilde{\mathbf{x}}_0 + \sqrt{\tau_n^2 - \alpha^2}\mathbf{x}$
8: $\quad \widetilde{\mathbf{x}}_0 \leftarrow \boldsymbol{F}_\theta(\mathbf{x}_{\tau_n}, \tau_n)$
9: **end for**
10: **Return:** $\widetilde{\mathbf{x}}_0$

---

## C EXPERIMENTAL DETAILS

We implemented all algorithms for training and evaluation using PyTorch (Paszke et al., 2019) 2.0.0, and all experiments were performed on 4 NVIDIA 3090 GPUs. For the Swiss roll, two moons, and CIFAR10 generation experiments, we set the source distribution of CCM/CCM-OT to a standard normal distribution. We use the RAdam (Liu et al., 2020) optimizer in training and keep the learning rate fixed. We set the weighted function $\lambda(\cdot)$ as the constant 1. In addition to the toy experiments, we used Mixed-Precision Training for both the CIFAR10 and AFHQ experiments to speed up and reduce memory. The detailed network structure, dataset preprocess and hyperparameter settings are described below.

### C.1 NETWORK ARTITECTURE

In the toy experiments, we just used a simple MLP as our model, this MLP has three layers containing 64 neurons per layer and added RELU activation functions after each layer, we concatenate anchor variables, e.g. time, on the input to the network. For the CIFAR-10 generation experiments, we refer to the network in earlier work on consistency modeling, please see Dhariwal & Nichol (2021) for a detailed description. For the experiments on AFHQ, we selected the DDPM++ network proposed in Song et al. (2020b) as the model.

We have introduced residual connectivity (Karras et al., 2022) to augment the model, specifically, the output of the model can be expressed as,

$$\boldsymbol{F}_\theta(\mathbf{x}, \tau) = c_{\text{skip}}(\tau)\mathbf{x} + c_{\text{out}}(\tau)f_\theta(c_{\text{in}}(\tau)\mathbf{x}, \tau) \tag{42}$$

where $f_\theta$ is the output of networks. $c_{\text{skip}}$, $c_{\text{out}}$ and $c_{\text{in}}$ represent scaling factors for residual, network output and input. They are each defined as follows,

$$c_{\text{skip}}(\tau) = \frac{\sigma_{\text{data}}^2}{(\tau - \alpha)^2 + \sigma_{\text{data}}^2}, \quad c_{\text{out}}(t) = \frac{\sigma_{\text{data}}(\tau - \alpha)}{\sqrt{\sigma_{\text{data}}^2 + \tau^2}}, \quad c_{\text{in}}(t) = \frac{1}{\sqrt{\sigma_{\text{data}}^2 + \tau^2}} \tag{43}$$

Here, $\alpha$ represents a small constant introduced to prevent numerical instability, while $\sigma_{\text{data}}$ is set to 0.5 in accordance with previous research (Karras et al., 2022). In practical terms, as outlined in Appendix B, we implement a step schedule denoted as $N(\cdot)$ and an EMA decay rate schedule represented by $\mu(\cdot)$ to optimize and enhance the training process. These schedules are defined as follows

$$N(k) = \left\lceil \sqrt{\frac{k}{k_{\text{max}}}\left((N_{\text{max}} + 1)^2 - N_{\text{min}}^2\right) + N_{\text{min}}^2} - 1 \right\rceil + 1$$

$$\mu(k) = \exp\left(\frac{N_{\text{min}}\log\mu_0}{N(k)}\right), \tag{44}$$

Here, we introduce two key parameters: $N_{\text{min}}$ and $N_{\text{max}}$, which represent the pre-defined minimum and maximum discrete steps, respectively. At the outset of training, $N(k)$ starts with a small value, leading to a substantial bias but limited variance, thereby facilitating rapid convergence. As the training progresses, $N(k)$ gradually increases, resulting in reduced deviation towards the end of training, while concurrently introducing a larger variance to enhance the final performance. Additionally, we employ the parameter $u_0$ to denote the initial value of EMA decay rate. Over the course of training, this rate progressively converges towards 1.

### C.2 DATASET DETAILS

**Toy Datasets**: We utilized Scikit-learn (Pedregosa et al., 2011) to generate a total of 50,000 samples per target distribution for training purposes and an additional 10,000 samples for testing. Prior to conducting our experiments, we applied Z-score normalization as a preprocessing step to all the generated samples. Our primary objective was to create distributions resembling a Swiss roll and two moons. To evaluate the quality of these generated distributions, we employed the Wasserstein-2 distance metric to quantify their similarity to the ground truth distribution. Additionally, we conducted experiments that involved transforming moons into rolls, allowing us to assess the adaptability of our CCM and CCM-OT models to arbitrary source distributions.

**CIFAR-10** (Krizhevsky et al., 2009): Images in CIFAR-10 were preprocessed by rescaling pixel values to the range [-1, 1], and horizontal flipping was applied for data augmentation. Training utilized the 50,000-image training dataset, with 50,000 images generated for FID calculation.

**CelebA** 64 (Yang et al., 2015): CelebA images underwent preprocessing steps including rescaling pixel values to [-1, 1], center cropping to $140 \times 140$, and resizing to $64 \times 64$. Models were trained on the training dataset, and FID was computed on a test set with 10,000 generated images.

**AFHQ** (Choi et al., 2020): The AFHQ dataset encompasses high-resolution images featuring animal faces across three distinct domains: cats, dogs, and wild animals, each displaying significant variability. The training dataset comprised 5,153 cat images, 4,739 dog images, and 4,738 wild animal images. Additionally, for each domain, there were 500 test images available. In preparation for our experiments, we rescaled the pixel values of all images to the range [-1, 1] and resized them to $256 \times 256$ pixels. We also applied horizontal flipping for data augmentation. Specifically, our experiments focused on translating between the *Cat→Dog* and *Wild→Dog* domains.

### C.3 TRAINING OVERHEAD ANALYSIS

Our proposed methods can be compared to the original DCM, potentially introducing additional computational overheads. We conducted a training time analysis on the CIFAR-10 dataset with a batch size of 256. Results showed a modest 1.8% increase in training time related to batch summation of vector fields for both the DCM-MS and PCM models. For the CCM-OT model, incorporating optimal transport for sample pairing led to a 3.3% rise in training time. Importantly, these operations occur independently of the network and, in our evaluation, did not significantly prolong computation times in practical training scenarios. Given the observed improvements in model performance, we consider the impact on overall computational efficiency to be reasonable.

## C.4 HYPERPARAMETERS

The hyperparameters for each model during training are shown below,

Table 4: Hyperparameter for toy experiments.

| Hyperparameter | Toy datasets | | |
| --- | --- | --- | --- |
| | DCM/DCM-MS | PCM | CCM/CCM-OT |
| Batch size | 512 | 512 | 512 |
| Learning rate | 1e-3 | 1e-3 | 1e-3 |
| $N_{min}$ | 2 | 2 | 2 |
| $N_{max}$ | 100 | 100 | 100 |
| $\mu_0$ | 0.95 | 0.95 | 0.95 |
| FP16 precision | No | No | No |
| Training iterations | 50K | 50K | 50K |
| $\alpha$ | 0.02 | 0.02 | 0.0001 |
| $T/r_{max}$ | 80.0 | 80.0 | 0.9999 |

Table 5: Hyperparameter for the Cifar-10 and AFHQ experiments.

| Hyperparameter | CIFAR-10 | | | AFHQ |
| --- | --- | --- | --- | --- |
| | DCM/DCM-MS | PCM | CCM/CCM-OT | CCM/CCM-OT |
| Batch size | 256 | 256 | 256 | 32 |
| Learning rate | 2e-4 | 2e-4 | 2e-4 | 1e-5 |
| $N_{min}$ | 2 | 2 | 2 | 2 |
| $N_{max}$ | 150 | 150 | 150 | 150 |
| $\mu_0$ | 0.95 | 0.95 | 0.95 | 0.95 |
| FP16 precision | Yes | Yes | Yes | Yes |
| Training iterations | 400K | 400K | 400K | 400K |
| $\alpha$ | 0.02 | 0.02 | 0.0001 | 0.0001 |
| $T/r_{max}$ | 80.0 | 80.0 | 0.9999 | 0.9999 |

Table 6: Hyperparameter for CelebA experiments.

| Hyperparameter | Toy datasets | | |
| --- | --- | --- | --- |
| | DCM/DCM-MS | PCM | CCM/CCM-OT |
| Batch size | 64 | 64 | 64 |
| Learning rate | 2e-4 | 2e-4 | 2e-4 |
| $N_{min}$ | 2 | 2 | 2 |
| $N_{max}$ | 150 | 150 | 150 |
| $\mu_0$ | 0.95 | 0.95 | 0.95 |
| FP16 precision | Yes | Yes | Yes |
| Training iterations | 200K | 200K | 200K |
| $\alpha$ | 0.02 | 0.02 | 0.0001 |
| $T/r_{max}$ | 80.0 | 80.0 | 0.9999 |

# D    EXTENDED SAMPLES

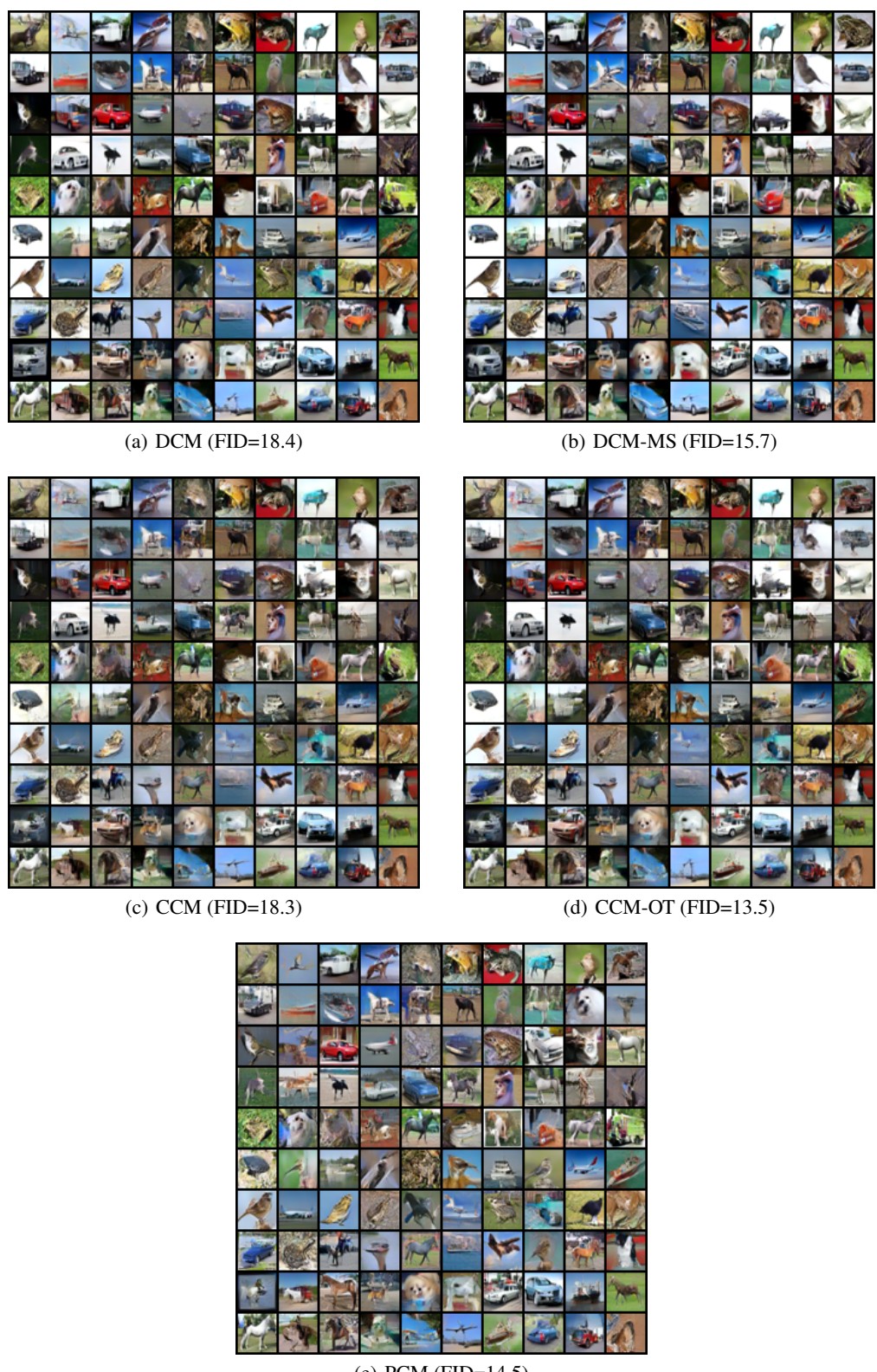

Figure 6:    Uncurated samples from CIFAR-10 $32 \times 32$. All corresponding samples use the same seed.

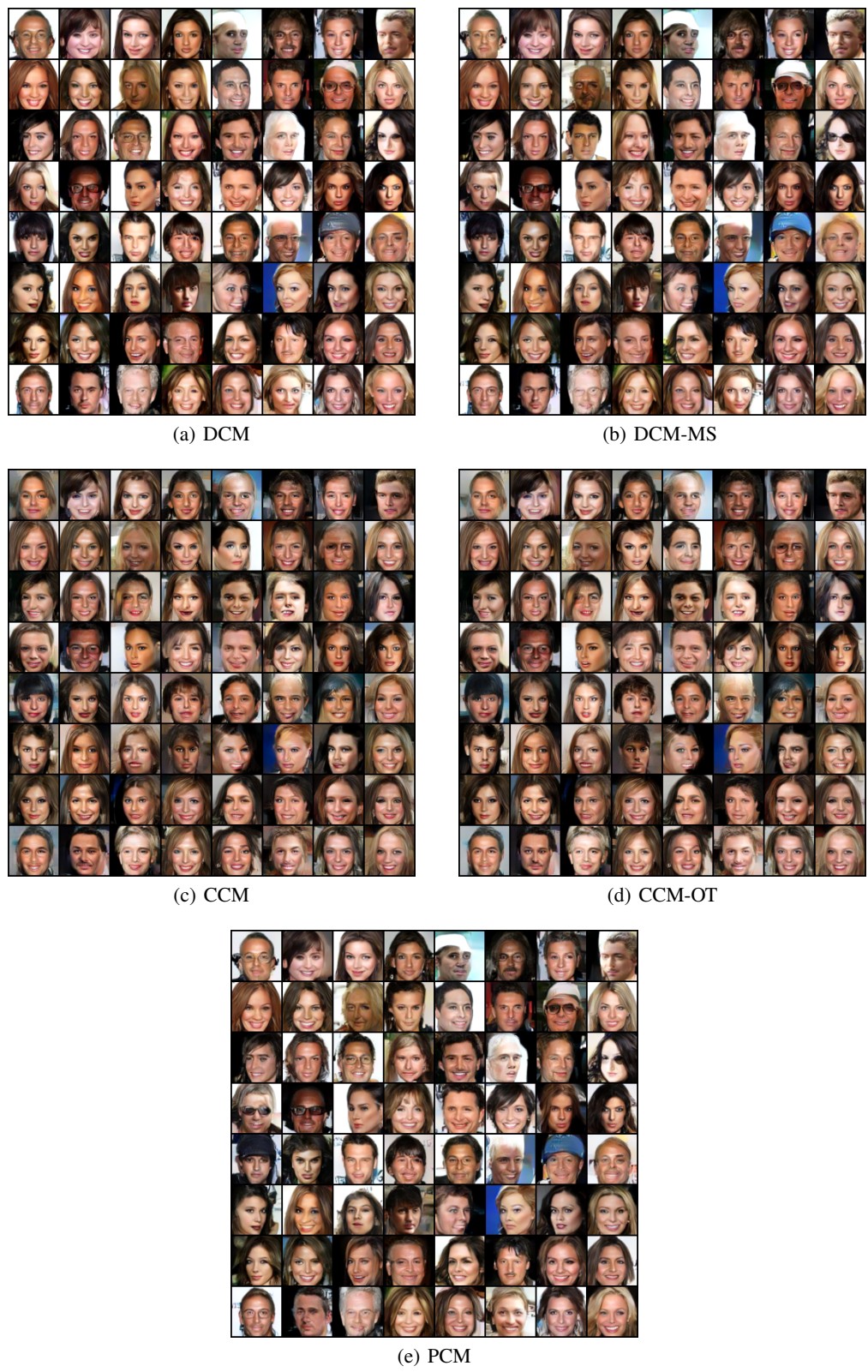

Figure 7: Uncurated samples from CelebA $64 \times 64$. All corresponding samples use the same seed.

$$Cat \rightarrow Dog$$

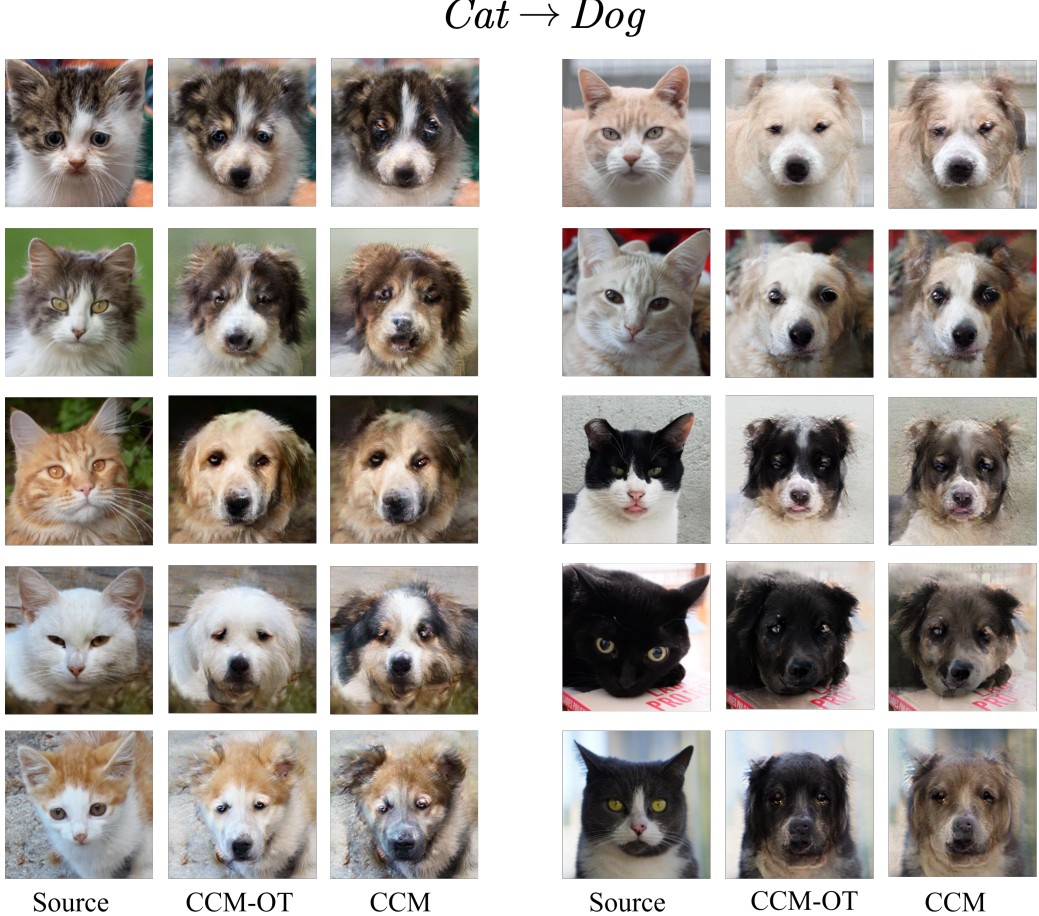

Figure 8: More qualitative results on *Cat→Dog* by CCM and CCM-OT.

$$Wild \rightarrow Dog$$

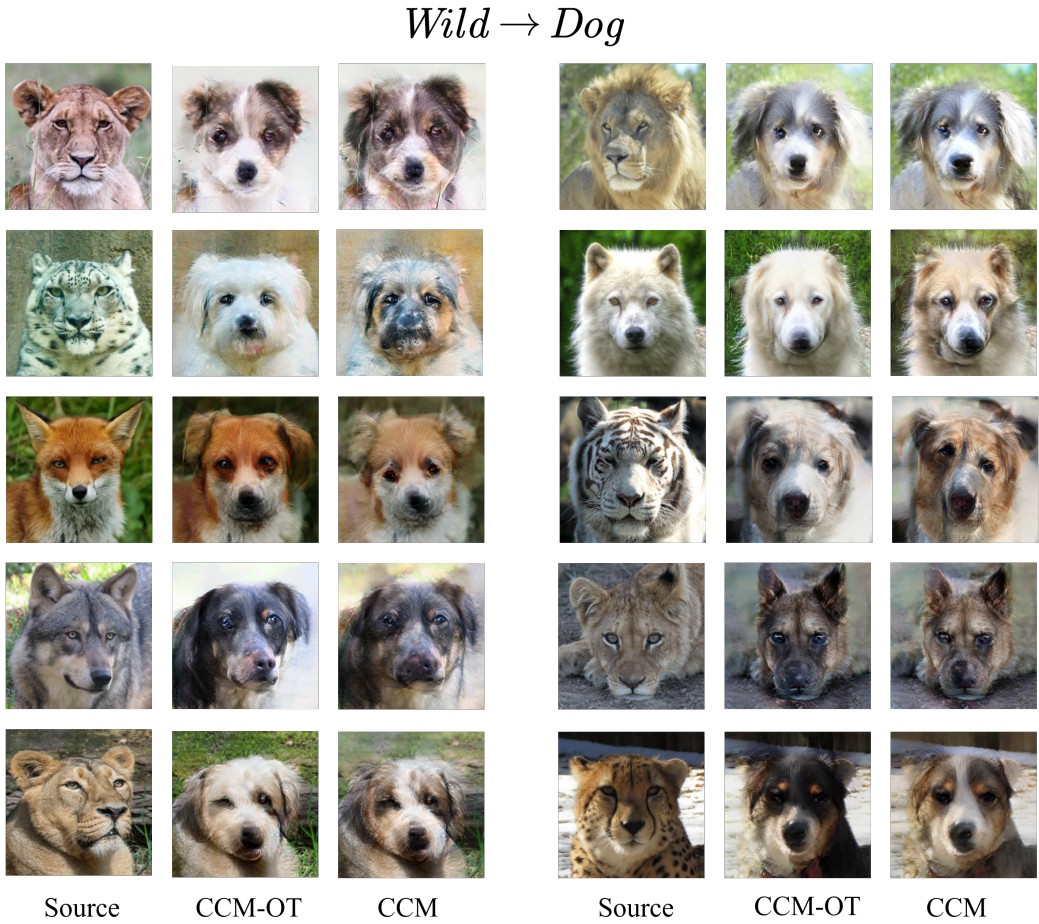

Figure 9: More qualitative results on *Wild →Dog* by CCM and CCM-OT.

