# OpenReview forum: "A Unified Framework for Consistency Generative Modeling"
_ICLR.cc/2024/Conference — ICLR 2024 Conference Withdrawn Submission_

### Official Review · Reviewer_Qpyg · 2023-11-01

**Soundness:** 3 good
**Presentation:** 3 good
**Contribution:** 3 good
**Rating:** 6
**Confidence:** 2

**Summary:**

This paper propose a way to unified three different views to formulate consistency model. This framework can inspires different design of consistency losses, and the paper demonstrate that such design can potentially lead to better consistency models. The paper is largely evaluated on three tasks, one is a toy 2D dataset, and then two image generations tasks. The results show that the new

**Strengths:**

- This theoretical framework that could unify multiple works (Poisson, Coupling, etc) and provide insight for developing new algorithm. To the best of my knowledge, this can be a useful contribution to the community. But I’m not an expert in generative model theory so I will need to defer this to other reviewers as well.

**Weaknesses:**

- Both proposed algorithm (PCM and CCM-OT) seems to require additional computes during training. For PCM, the weighted sum is computed through all x_i in the batch and for CCM-OT, the optimal transport is computed among the batch. These operations are not scaling very trivially with the batch-size, while to the best of my knowledge, consistency model seems to work better with larger batch size (e.g. 512 and in this paper case 256).

**Questions:**

- the final algorithm is not very clear to me by reading the main paper, there are several tricks proposed. It would be great to have the algorithm written out in the main paper rather than in the appendix.

---

> ### Author Response · Authors · 2023-11-18
> **Thank you for your review and suggestions**
>
> Dear Reviewer Qpyg,
>
> Thank you for recognizing our contribution to our work and the constructive feedback. Here is our response to your concerns:
> *******************************************
>
> **Q: Both proposed algorithm (PCM and CCM-OT) seems to require additional computes during training. For PCM, the weighted sum is computed through all $x_i$ in the batch and for CCM-OT, the optimal transport is computed among the batch. These operations are not scaling very trivially with the batch-size, while to the best of my knowledge, consistency model seems to work better with larger batch size (e.g. 512 and in this paper case 256).**
>
> **A:** Thank you for highlighting the potential increase in computational complexity introduced by the proposed enhancements in our paper. To address this concern, we conducted an analysis of the training time on the CIFAR-10 dataset with batchsize 256:
>
> 1) **Batch Weighted Summation.** We observed a modest 1.8% increase in training time related to batch summation for both the DCM-MS and PCM models.
>
> 2) **Optimal Transport.** In the case of the CCM-OT model, which incorporates optimal transport for sample pairing, we noted a 3.3% rise in training time associated with optimal transmission.
>
> We noted that these operations all occur independently of the network and, in our evaluation, did not result in significantly prolonged computation times for practical training scenarios. Given the improvements in model performance, we believe the impact on overall computational efficiency is reasonable.
>
> Thank you again for your concerns, and we will include a discussion of computational overhead in the revised version.
> *******************************************
> **Q: The final algorithm is not very clear to me by reading the main paper, there are several tricks proposed. It would be great to have the algorithm written out in the main paper rather than in the appendix.**
>
> **A:** We appreciate your thoughtful feedback and recognize the importance of providing a clear understanding of our proposed algorithm. Considering the constraints of page limitations, we regret that we are unable to present the complete training and sampling algorithm in the main text.
>
> To address this concern and enhance the accessibility of our work, we will incorporate a jump link in the main paper, directing readers to the corresponding pseudocode in the appendix. This approach will enable readers to seamlessly navigate between the main text and the detailed algorithmic representations for each model, ensuring a more comprehensive understanding of the proposed methodology.
>
> We look forward to your continued insights and hope that this adjustment enhances the clarity and accessibility of our work.

---

> ### Author Response · Authors · 2023-11-22
> **Thank you for your time and any follow up questions?**
>
> Dear Reviewer Qpyg,
>
> Towards the end of the discussion phase, we are optimistic that our response has effectively addressed the queries raised. We eagerly await your feedback to ascertain if our reply adequately resolves any concerns you may have or if further clarification is required.
>
> Thank you for your time and consideration.
>
> Sincerely,
>
> Paper 3337 Authors

---

### Official Review · Reviewer_drKn · 2023-11-01

**Soundness:** 3 good
**Presentation:** 3 good
**Contribution:** 3 good
**Rating:** 6
**Confidence:** 3

**Summary:**

This paper introduces a unified and comprehensive framework for consistency generative modeling. In particular, it introduces two novel models: Poisson Consistency Models (PCMs) and Coupling Consistency Models (CCMs), which extend the prior distribution of latent variables beyond the Gaussian form. Additionally, it incorporates optimal transport (OT) to improve the performance of these models further. Through empirical experiments, it demonstrates the effectiveness of the proposed framework across a range of generative tasks.

**Strengths:**

This paper generalizes consistency models beyond the Gaussian form. In particular, it introduces two novel models: Poisson Consistency Models (PCMs) and Coupling Consistency Models (CCMs) with theoretical analysis. And comprehensive experiments show advantages compared to other diffusion models on both synthetic and real-world datasets, such as unconditional generation of CIFAR-10 and unpaired image-to- image translation (I2I) using AFHQ.

**Weaknesses:**

1. the notation is a little confusing, which makes the paper hard to follow.
2. The  reverse diffusion process with gaussian noise can reconstruct the original clear image from the noisy version step by step, it would be better to show some figures using PCMs and CCMs.

**Questions:**

From Eq. 22 and 23, it shows that the continuity equation holds with or without condition z, then what is the physical meaning for z?

---

> ### Author Response · Authors · 2023-11-18
> **Thank you for your review and suggestions**
>
> Dear Reviewer drKn,
>
> Thank you for the detailed review and thoughtful feedback. Below we address specific questions.
> ***************************************
> **Q: the notation is a little confusing, which makes the paper hard to follow.**
>
> **A:** I apologize for any confusion caused by our current notation choices. In the revised version, we are actively working to enhance the use of symbols for a clearer and more easily understandable presentation. If you have specific suggestions or examples where the notation can be improved, please feel free to provide them, as your input is invaluable in refining our paper for better comprehension.
> ***************************************
> **Q: The reverse diffusion process with gaussian noise can reconstruct the original clear image from the noisy version step by step, it would be better to show some figures using PCMs and CCMs.**
>
> **A:** Thank you for raising this point. While our consistency modeling framework shares similarities with the diffusion model, it distinguishes itself by directly recovering the image from noise through a single-step mapping, bypassing intermediate stages.
>
> ********************************************
> **Q: From Eq. 22 and 23, it shows that the continuity equation holds with or without condition z, then what is the physical meaning for $z$?**
>
> **A:**  I appreciate your insightful question. You are correct; the continuity equation describes the relationship between the probability path and the velocity vector field, and its establishment does not depend on $z$. In this context, $z$ lacks a clear physical meaning; we use it as a tool to indirectly construct the desired probability path for consistency training.

---

> ### Author Response · Authors · 2023-11-22
> **Thank you for your time and any follow up questions?**
>
> Dear Reviewer drKn,
>
> Towards the end of the discussion phase, we are optimistic that our response has effectively addressed the queries raised. We eagerly await your feedback to ascertain if our reply adequately resolves any concerns you may have or if further clarification is required.
>
> Thank you for your time and consideration.
>
> Sincerely,
>
> Paper 3337 Authors

---

### Official Review · Reviewer_i9Sm · 2023-11-01

**Soundness:** 2 fair
**Presentation:** 3 good
**Contribution:** 1 poor
**Rating:** 3
**Confidence:** 4

**Summary:**

This work proposed a generalized formulation of the Consistency Model [CM, Song et al. 2023], a recent advanced in diffusion generative modeling that enable few number of function evaluation (usually only 1 or 2 NFEs)-sampling but high-quality samples. The authors show that CM can be generalized into a general form with affine probability path that admits a velocity field. This results in an equivalent form continuity equation that describes the dynamic of the probabiliy density path. Equip with this general form, the authors proposed two additional extension of CM, called Poisson consistency model (PCM), based on formulation of the probability path showed in [Xu et al 2022], and Coupling consistency model (CCM), based on the linear probability path of flow matching framework [Lipman et al 2023, Liu et al 2023, Albergo & Vanden-Ejinden 2023]. The authors demonstrate the effectiveness of their proposed methodologies with numerical experiments on unconditional image generation task (CIFAR10) and unpaired image-to-image translation (AFHQ Cat-Dog/Wild-Dog).



Song, Y., Dhariwal, P., Chen, M. &amp; Sutskever, I.. (2023). Consistency Models. In Proceedings of Machine Learning Research202:32211-32252 Available from https://proceedings.mlr.press/v202/song23a.html.

Xu, Y., Liu, Z., Tegmark, M., & Jaakkola, T. (2022). Poisson flow generative models. Advances in Neural Information Processing Systems, 35, 16782-16795.

Yaron Lipman, Ricky TQ Chen, Heli Ben-Hamu, Maximilian Nickel, and Matt Le. Flow matching for generative modeling. arXiv preprint arXiv:2210.02747, ICLR 2023.

Liu, Xingchao, Chengyue Gong, and Qiang Liu. "Flow straight and fast: Learning to generate and transfer data with rectified flow." arXiv preprint arXiv:2209.03003. ICLR 2023.

Albergo, Michael Samuel, and Eric Vanden-Eijnden. "Building Normalizing Flows with Stochastic Interpolants." In The Eleventh International Conference on Learning Representations, 2023.

**Strengths:**

Well written and structured paper that is easy to pass through. The main goal of improving and generalizing consistency model framework is well-motivated.

**Weaknesses:**

I have two main concerns for this paper.

1. **Question mark on novelty:** the idea of writing continuity equation and probability path is not new, and many derivations  from the generalized CM in this paper are followed exactly from [Lipman et al 2023] on flow matching generative models. The formulation of Couping Consistency Model (CCM) -- section 3.4, which is this work's most well-performed framework empirically, the authors again borrowed heavily from previous work of [Lipman et al. 2023] with the linear path $x_t = tx_1 + (1-t)x_0$. Equation (20) and the idea of CCM-OT on learning probability path with joint distribution to reduce training loss variance is straightforward taken from Multisample Flow matching paper [Pooladian et al. 2023, Section 3], but surprisingly there was no mention of this citation around this equation.

2. **Question mark on the results of empirical evaluation:** although the results provided with 1 NFE sampling for CIFAR10 in table 2 and 3 showed that CCM-OT outperformed original CM (denoted DCM in this paper) of [Song et al. 2023], I do not understand where the authors got the FID score of 18.4 for DCM to begin with. In [Song et al. 2023], it is clearly stated that their model reached 1-NFE FID 3.55 & 8.70 with distillation and without distillation, respectively. Therefore, I urge the authors provide a clear explanation on the discrepancy between the reported numbers of the baseline. Moreover, DCM also provided more extensive experiments with ImageNet 64x64 and LSUN Bedroom 256x256, and although I understand the lack of computational resources, I advise the authors to provide results on more datasets for better understanding the gains of their proposed methods compared with baselines, which should also included distillation techniques such as Progressive distillation [Salisman & Ho 2022].

Yaron Lipman, Ricky TQ Chen, Heli Ben-Hamu, Maximilian Nickel, and Matt Le. Flow matching for generative modeling. arXiv preprint arXiv:2210.02747, ICLR 2023.

Pooladian, Aram-Alexandre, Heli Ben-Hamu, Carles Domingo-Enrich, Brandon Amos, Yaron Lipman, and Ricky Chen. "Multisample flow matching: Straightening flows with minibatch couplings." arXiv preprint arXiv:2304.14772 (2023).

Salimans, T. and Ho, J. Progressive distillation for fast sampling of diffusion models. In International Conference on Learning Representations, 2022. URL https://openreview.net/forum?id=TIdIXIpzhoI

**Questions:**

See weaknesses.

---

> ### Author Response · Authors · 2023-11-18
> **Thank you for your review and suggestions**
>
> Dear Reviewer i9Sm,
>
> We sincerely appreciate your positive comments and feedback, as well as the suggestions for improvement. In response to your questions, we have prepared the answers below:
> **************************************************
> **Q: Many derivations from the generalized CM in this paper are followed exactly from [1] on flow matching generative models. The formulation of Couping Consistency Model (CCM) -- section 3.4, which is this work's most well-performed framework empirically, the authors again borrowed heavily from previous work of [1] with the linear path .**
>
> **A:** Thank you for highlighting this observation. In Section 3.1, we explicitly acknowledge that our approach to constructing probabilistic paths draws inspiration from recent advances in Flow-based models [1]. It is essential to recognize, however, that our primary focus diverges significantly.
>
> Firstly, our major contribution lies in unveiling the intrinsic relationship between probabilistic paths and consistency models (CMs). This unique perspective enables us to approach consistency training from a foundational angle, potentially catalyzing the development of innovative single-step generative models. The introduction of the linear path into consistency models facilitates a distinct single-step Im2Im translation. In contrast, [1] typically employs around 100~200 steps, underscoring the non-trivial nature of our contribution.
>
> Secondly, it's imperative to emphasize that the conclusions in [1] are limited to Gaussian paths. In our work, we generalize these conclusions to a broader form, a crucial step for incorporating Poisson Consistency Models (PCMs) into our proposed framework. This generalization significantly enhances the adaptability and applicability of our approach beyond the constraints of Gaussian paths established by [1].
> ******************************************************
>
> **Q:Equation (20) and the idea of CCM-OT on learning probability path with joint distribution to reduce training loss variance is straightforward taken from Multisample Flow matching paper [2]**
>
> **A:** We appreciate your insightful observation and acknowledge the connection between our CCM-OT and optimal transport flow matching in [2]. In our related section (Section 4), we have cited and expounded upon the relevance of their work, underscoring the differences in our motivation.
>
> While [2] demonstrates the efficacy of coupling distributions in reducing the variance of the flow matching target, we place a specific emphasis on refining the estimation of vector fields within our proposed framework, constituting a unique contribution in the realm of CMs. To the best of our knowledge, there exists no prior work proposing the link of optimal transport and CMs. Consequently, our work fills a significant gap in this domain, introducing a novel application of optimal transport principles to elevate the performance of CMs.
>
> We are grateful for your suggestion, and as you recommended, we will refer back to the work [2] in the corresponding section to ensure that readers gain a comprehensive understanding of our contributions.
> ***********************************************
>
> *[1] Lipman Y, Chen R T Q, Ben-Hamu H, et al. Flow matching for generative modeling[J]. arXiv preprint arXiv:2210.02747, 2022.*
>
>
> *[2] Pooladian A A, Ben-Hamu H, Domingo-Enrich C, et al. Multisample flow matching: Straightening flows with minibatch couplings[J]. arXiv preprint arXiv:2304.14772, 2023.*

---

> ### Author Response · Authors · 2023-11-18
> **Part II. Rebuttal**
>
> **Q:I urge the authors provide a clear explanation on the discrepancy between the reported numbers of the baseline.**
>
> **A:** Thank you for bringing attention to the differences in our reported metrics compared to [1]. Specifically, we meticulously replicated the work of [1], maintaining consistency in **time discretization, network design, learning rate, moving average rate, loss computation, and other key aspects**. However, due to computational constraints, we made necessary adjustments, including reducing the batch size from 512 to 256, and the number of iterations from 800K to 400K. Additionally, we implemented mixed-precision training for further acceleration.
>
> Despite these modifications, we want to emphasize our comparisons are fair. All models were configured with identical experimental hyperparameters. We believe that the primary source of the observed discrepancies may be attributed to the use of a smaller batchsize in our implementation. We will further elucidate our experimental setup and discrepancy compared with [1] in the revised version.
> ****************************************************
>
> **Q:I advise the authors to provide results on more datasets for better understanding the gains of their proposed methods compared with baselines.**
>
> **A:** We appreciate your valuable feedback and fully acknowledge the merit of conducting evaluations on a broader set of datasets. However, it's worth noting that datasets like Imagnet and LSUN Bedroom, used by [1], involved training a consistency model on 64 A100 GPUs clusters—such computational resources are beyond our current capacity.
>
> Despite these constraints, we have extended our evaluation to include another widely recognized image-generated benchmark dataset, CelebA. All methods were trained with a batch size of 64 and updated for 200K iterations and the results are as follows,
>
> | method | | FID | |
> |  ----  | ----  | ----  | ----  |
> | | NFE=1|NFE=2|NFE=5|
> | DCM | 41.0 | 27.3 | 25.3 |
> | DCM-MS | 38.3 | 22.2 | 18.2|
> | PCM | 32.9 | **17.8** | **16.1** |
> | CCM-OT | **30.8** | 28.0| 25.5|
>
> We observed consistent improvements with our proposed method, reinforcing its efficacy. We understand the importance of comprehensive evaluations and hope that these additional results contribute to a better understanding of the strengths of our approach.
> *****************************************************************
> **Q:Comparison with distillation technology such as Progressive distillation [2]**
>
> **A:** We appreciate the suggestion to compare our approach with distillation technologies like Progressive Distillation [4]. However, it's important to note that our primary focus in this paper is on the consistency training introduced by [1]. Our objective is to achieve a single-step generative model directly, without relying on pre-trained score networks.
>
> Unlike distillation approaches based on score networks, our consistency framework can be positioned as an independent family of generative models that are computationally resource-friendly.
> ****************************************************
> *[1] Song Y, Dhariwal P, Chen M, et al. Consistency models[J]. 2023.*
>
> *[2] Salimans T, Ho J. Progressive distillation for fast sampling of diffusion models[J]. arXiv preprint arXiv:2202.00512, 2022.*

---

> ### Comment · Reviewer_i9Sm · 2023-11-21
> **Thank you for the rebuttal, but I am keeping my original opinion.**
>
> I have read the rebuttal of the author. I appreciate the efforts that the authors put into it. However, I don't think my two major concerns are addressed.
>
> 1. Unfair comparison: if I am not wrong, the authors cited a misleading information about computational requirement for consistency model [1], as I could not find the line in that paper that explicitly said they use exactly 64 A100 GPUs (which I agree is a lot), but only state a cluster of A100 GPUs. I would be happy to be clarified about this. For a check, I could also suggest the authors share their code so I and other reviewers can check the reproducibility of their experiments.
>
> 2. Novelty: I restate the fact that flow matching and diffusion model (more exactly probability flow) are in fact very closely related to each other (with both sharing the same affine probability path) has been pointed out by flow matching papers [2,3]. Combining them with consistency model for diffusion probabiliy path in [1] makes it become the CCM in this paper, and adding an layer of OT coupling that has been introduced in [4] to make it become CCM-OT. I do agree that the framework the authors presented is new, but is still a combination of multiple existing ideas, which makes the novelty limited.
>
>
> [1] Song Y, Dhariwal P, Chen M, et al. Consistency models. ICML 2023.
>
> [2] Lipman Y, Chen R T Q, Ben-Hamu H, et al. Flow matching for generative modeling[J]. arXiv preprint arXiv:2210.02747, 2022.
>
> [3] Liu, Xingchao, Chengyue Gong, and Qiang Liu. "Flow straight and fast: Learning to generate and transfer data with rectified flow." arXiv preprint arXiv:2209.03003. ICLR 2023.
>
> [4]  Pooladian, Aram-Alexandre, Heli Ben-Hamu, Carles Domingo-Enrich, Brandon Amos, Yaron Lipman, and Ricky Chen. "Multisample flow matching: Straightening flows with minibatch couplings." arXiv preprint arXiv:2304.14772 (2023).

---

> > ### Author Response · Authors · 2023-11-21
> > **Further clarification of experiments and contributions**
> >
> > We appreciate the time and effort you've invested in reviewing our work. We address your concerns below.
> > *****************************************
> > **Computational Resources:** In [1], they shows the number of GPU for training models in Table 3 of their Appendix C. There are 8 GPUs used for CIFAR-10 and 64 for ImageNet and LSUN bedroom.
> > *****************************************
> >
> > **Code Availability:** In response to the recommendation to provide an code repository, we now offer an anonymous repository (https://anonymous.4open.science/r/UniCM-18A0/), allowing both the reviewer and the wider community to scrutinize our implementation for further verification.
> > *****************************************
> >
> > **Novelty clarification:** We appreciate your insightful comments regarding the intersection of flow matching and diffusion modeling. We acknowledge the inherent connections between these domains and recognize that they can be integrated under specific circumstances. However, we wish to underscore the distinctiveness of our work, which centers on introducing a unified perspective within another independent family of generative models—consistency models.
> >
> > Our proposed CCM/CCM-OT, drawing inspiration from the linear probability path [2,3], should be viewed as a specific instantiation within our comprehensive framework. We are confident that our framework constitutes a foundational contribution, establishing a robust basis for future advancements in the development and design of novel consistency models. This represents our primary contribution to the generative modeling field.
> > *****************************************
> > *[1] Song Y, Dhariwal P, Chen M, et al. Consistency models. ICML 2023.*
> >
> > *[2] Lipman Y, Chen R T Q, Ben-Hamu H, et al. Flow matching for generative modeling[J]. arXiv preprint arXiv:2210.02747, 2022.*
> >
> > *[3] Liu, Xingchao, Chengyue Gong, and Qiang Liu. "Flow straight and fast: Learning to generate and transfer data with rectified flow." arXiv preprint arXiv:2209.03003. ICLR 2023.*

---

> > > ### Comment · Reviewer_i9Sm · 2023-11-22
> > > **Re: Further clarification**
> > >
> > > Thank you for publishing the code and further clarification. Could you quickly clarify why there are difference between variance schedulers config (sigma_min, sigma_max) of the two model type `DCM` and `CCM/CCM-OT`?

---

> ### Author Response · Authors · 2023-11-22
> **variance schedulers**
>
> Thank you for your question. sigma_min and sigma_max in the code represent the range of the anchor variables $t$ in our paper. For DCM, we followed the settings of [1] with sigma_min=$0.02$, sigma_max=$80$, and $t\in [0.02, 80]$.  For CCM/CCM-OT, we compute the interpolation to sample from the conditional probability paths, i.e. $tx_1+(1-t) x_0$, where $t\in [0, 1]$. In order to prevent numerical errors, we set it as sigma_min=$0.0001$, sigma_max=$0.9999$.
>
>
> *[1] Song Y, Dhariwal P, Chen M, et al. Consistency models. ICML 2023.*

---

> > ### Comment · Reviewer_i9Sm · 2023-11-22
> > **Re: Further clarification**
> >
> > Hi, what I meant is that would you be able to clarify that this discrepancy in setting variance schedulers would not results in discrepancy of the performance of the two methods, and not just because the difference in probability path formulation? Thanks.

---

> ### Author Response · Authors · 2023-11-22
> **variance schedulers**
>
> We are sorry that we initially misunderstood your question. We think that these two variance schedulers are similar to the two setups of variance preservation and variance explosion in the field of diffusion model, both of them aim at corrupting the original image signals by a prior distribution, and we believe that both of them do not make a significant difference on the model performance, as can be seen from our experimental results on DCM and CCM. We hope our answers address your concerns.

---

> > ### Comment · Reviewer_i9Sm · 2023-11-22
> > **Re: variance schedulers**
> >
> > Hi, I am asking this question because as far as I am aware, using different variance schedulers __do__ make big diffrences in terms of performance of the models in both sampling and training (in this case the authors said VP and VE SDE), see e.g. simple diffusion paper [1] for stochastic path and EDM paper for ode deterministic path [2, Section 3]. There are enough discrepancies in the performance regarding FID between DCM (the baseline) and CCM, for example on NFE=2 on CIFAR10, or CelebA-64 where actually CCM is performing better (and I suspect it will be bigger if we increase the resolution of the datasets).
> >
> > I appreciate the authors' intuitive answer, but I am keeping my original evaluation.
> >
> > [1] Hoogeboom, E., Heek, J. &amp; Salimans, T.. (2023). simple diffusion: End-to-end diffusion for high resolution images. Proceedings of the 40th International Conference on Machine Learning 202:13213-13232.
> >
> > [2] Karras, T., Aittala, M., Aila, T., & Laine, S. (2022). Elucidating the design space of diffusion-based generative models. Advances in Neural Information Processing Systems, 35, 26565-26577.

---

> ### Author Response · Authors · 2023-11-22
> **Thank you for your time and any follow up questions?**
>
> Dear Reviewer i9Sm,
>
> Towards the end of the discussion phase, we are optimistic that our response has effectively addressed the queries raised. We eagerly await your feedback to ascertain if our reply adequately resolves any concerns you may have or if further clarification is required.
> Thank you for your time and consideration.
>
> Sincerely,
>
> Paper 3337 Authors

---

### Official Review · Reviewer_Dtgk · 2023-11-01

**Soundness:** 2 fair
**Presentation:** 2 fair
**Contribution:** 2 fair
**Rating:** 5
**Confidence:** 3

**Summary:**

This paper proposes a unified framework for consistency generative modeling by introducing two models: Poisson Consistency Models(PCMs) and Coupling Consistency Models(CCMs). The overall training pipeline stays similar to the original Consistency Models. The PCMs attempt to replace the Gaussian distribution of the latent variables in the diffusion models with the Poisson distribution while the CCMs introduce a tuple of random variables to replace the Gaussian distributions. The paper also utilizes Opticmal Transport to further improve the performance of CCMs. Results are shown on a toy dataset and CIFAR-10.

**Strengths:**

1. Speeding up the diffusion models is an important problem and consistency training has been shown to be an effective way to achieve this.
2. Results on the toy experiments and CIFAR-10 show that the proposed methods improve over the baselines regarding of FID.

**Weaknesses:**

1. The evaluation is weak. The paper only conducts quantitative comparisons on a toy dataset and CIFAR-10.
2. As in the Consistency Models paper, the consistency distillation method generally performs better than consistency training. However, the paper does not provide any results or comparisons for this.

**Questions:**

The key contribution that goes beyond the gaussian distribution by replacing the gaussian distribution with either poisson distribution or a joint distribution seems to not specific to the training of consistency models but also the vanilla diffusion models. Could the authors clarify more for on this?

---

> ### Author Response · Authors · 2023-11-18
> **Thank you for your review and suggestions**
>
> Dear Reviewer Dtgk,
>
> Thank you for the detailed review and thoughtful feedback. Below we address specific questions.
> ************************
> **Q: The evaluation is weak. The paper only conducts quantitative comparisons on a toy dataset and CIFAR-10.**
>
> **A:** We appreciate your pointer and acknowledge the need for a more evaluation. In response, we will include another benchmark dataset, Celeba64 $ \times $64, in the revised version. All methods were trained with a batch size of 64 and updated for 200K iterations, yielding the following results:
>
> | method | | FID | |
> |  ----  | ----  | ----  | ----  |
> | | NFE=1|NFE=2|NFE=5|
> | DCM | 41.0 | 27.3 | 25.3 |
> | DCM-MS | 38.3 | 22.2 | 18.2|
> | PCM | 32.9 | **17.8** | **16.1** |
> | CCM-OT | **30.8** | 28.0| 25.5|
>
> The result further highlights the advantages of our proposed algorithm over the original DCM, affirming the effectiveness of our framework. We commit to promptly include these tables and visualizations in the revised version.
>
> It is crucial to emphasize that our work extends the modeling capabilities of consistency models (CMs) [1]. In addition to unconditional generation, we introduced a unique single-step unpaired im2im experiment on AFHQ, addressing limitations in previous CMs related to the diffusion process.
> ************************
> **Q: As in the Consistency Models paper, the consistency distillation method generally performs better than consistency training. However, the paper does not provide any results or comparisons for this.**
>
> **A:** Thank you for bringing up this point. Our framework is designed to establish a self-sufficient and self-reliant approach for consistency generative modeling, without dependence on pre-trained score networks.
>
> While we do mention consistency distillation in our work, it was not the primary focus for several reasons:
>
> 1) Incorporating consistency distillation necessitates an additional pre-training step, introducing computational overhead and somewhat deviating from our original design philosophy for this framework.
>
> 2) The effectiveness of consistency distillation is inherently depend on the quality of the pre-trained model used. Concurrent research [2] has shown that independent consistency modeling can achieve significant improvements through various advanced training tricks, potentially surpassing the performance of consistency distillation. We believe our framework can also benefit from these strategies.
> *******************************************************
>
> **Q: The key contribution that goes beyond the gaussian distribution by replacing the gaussian distribution with either poisson distribution or a joint distribution seems to not specific to the training of consistency models but also the vanilla diffusion models. Could the authors clarify more for on this?**
>
> **A:** I appreciate the reviewer's thoughtful question. Our work indeed is inspired by prior efforts that have explored modifications to the Gaussian distribution in vanilla diffusion models, such as the Poisson flow [3] and Flow matching [4]. However, it is crucial to highlight that our contribution extends beyond a simple replacement of distributions.
>
> Our distinctive contribution lies in establishing a profound connection between the self-consistency property inherent in Consistency Models (CMs) and the probabilistic paths. This profound connection not only enhances our understanding of CMs but also facilitates the design of more effective single-step generative models.
>
> *********************************************************
> *[1] Song Y, Dhariwal P, Chen M, et al. Consistency models[J]. 2023.*
>
> *[2] Song Y, Dhariwal P. Improved Techniques for Training Consistency Models[J]. arXiv preprint arXiv:2310.14189, 2023.*
>
> *[3] Xu Y, Liu Z, Tegmark M, et al. Poisson flow generative models[J]. Advances in Neural Information Processing Systems, 2022, 35: 16782-16795.*
>
> *[4] Lipman Y, Chen R T Q, Ben-Hamu H, et al. Flow matching for generative modeling[J]. arXiv preprint arXiv:2210.02747, 2022.*

---

> ### Author Response · Authors · 2023-11-22
> **Thank you for the review and any follow up questions?**
>
> Dear Reviewer Dtgk,
>
> Towards the end of the discussion phase, we are optimistic that our response has effectively addressed the queries raised. We eagerly await your feedback to ascertain if our reply adequately resolves any concerns you may have or if further clarification is required.
>
> Thank you for your time and consideration.
>
> Sincerely,
>
> Paper 3337 Authors

---

### Author Response · Authors · 2023-11-19
**A summary of updates**

Dear Reviewers,

We would like to thank all reviewers for their constructive feedback. We have revised our paper according to these comments. Major revisions are highlighted in **red** in the new version. Specifically, we have modified the following main points:
**********************************************************
**Additional experiments and setup clarifications.**
To bolster the robustness of our evaluation, we conducted experiments on the CelebA dataset. Quantitative results are now presented in a modified Table 3, accompanied by visualization results in Appendix D. To elucidate any discrepancies reported with the results in [1], we have included a detailed discussion in the experimental section.
**********************************************************
**Citation of related work.**
In response to Reviewer i9Sm's inquiry regarding the novelty of our work, we have incorporated citations to relevant works ([2,3]) in Section 3.4. Further clarification of the distinctions and relationships between our work and these references has been expounded upon in the related work section.

**********************************************************
**Clarity and readability of the proposed algorithm.**
Addressing the suggestions from Reviewers drKn and Qpyg to enhance the readability of the algorithms, we have included hyperlinks to the pseudo-code of the algorithms in the appendices. Additionally, a discussion on computational overheads has been added in Appendix C.3 for a more comprehensive understanding.
***********************************************
*[1] Song Y, Dhariwal P, Chen M, et al. Consistency models[J]. 2023.*

*[2] Lipman Y, Chen R T Q, Ben-Hamu H, et al. Flow matching for generative modeling[J]. arXiv preprint arXiv:2210.02747, 2022.*

*[3] Pooladian A A, Ben-Hamu H, Domingo-Enrich C, et al. Multisample flow matching: Straightening flows with minibatch couplings[J]. arXiv preprint arXiv:2304.14772, 2023.*
*****************************************
We hope our response is sufficient to address your questions. Please kindly let us know if you have any additional concerns.

Paper 3337 Authors

---

### Comment · Area_Chair_dVh2 · 2023-11-21

Dear Reviewers:

In light of the disagreements regarding the paper's merits, I encourage you to reassess the paper, taking into consideration the provided rebuttals and revisions. Please inform the authors of any changes to your opinions, whether they have improved or worsened.

Thanks,

AC

---

### Meta-Review · Area_Chair_dVh2 · 2023-12-08

**Metareview:**

This paper presents a unified framework for consistency generative modeling, introducing Poisson Consistency Models (PCMs) and Coupling Consistency Models (CCMs) that extend the distribution of latent variables beyond Gaussian forms. The incorporation of Optimal Transport enhances their performance, as demonstrated in empirical experiments across generative tasks such as unconditional image generation (CIFAR10) and unpaired image-to-image translation (AFHQ Cat-Dog/Wild-Dog).

Regarding concerns about the comparative analysis, the AC shares reservations about the reported consistency model results on CIFAR10, noting significant deviations from the original paper. The attributed cause, a reduced batch size of 256 instead of 512, is questioned, with a suggestion that employing gradient accumulation on four RTX 3090 GPUs could potentially address any out-of-memory challenges associated with a larger batch size.

Furthermore, the manuscript is criticized for lacking clarity regarding its original contributions. The intertwined presentation of various elements hinders a clear delineation of distinct contributions, impeding a comprehensive understanding of the paper's novel insights.

**Justification For Why Not Higher Score:**

In light of these identified concerns, the AC contends that further revisions are imperative before the paper attains the level of refinement necessary for publication in ICLR.

**Justification For Why Not Lower Score:**

N/A

---

### Decision · Program_Chairs · 2024-01-16

Reject